# Overcoming low initial coulombic efficiencies of Si anodes through prelithiation in all-solid-state batteries

So-Yeon Ham [1], Elias Sebti [2], Ashley Cronk[1], Tyler Pennebaker [2], Grayson Deysher[1], Yu-Ting Chen[1], Jin An Sam Oh [3], Jeong Beom Lee [4], Min Sang Song[4], Phillip Ridley[5], Darren H. S. Tan[5], Raphaële J. Clément [2], Jihyun Jang [5,6] ✉ & Ying Shirley Meng [5,7] ✉

All-solid-state batteries using Si as the anode have shown promising performance without continual solid-electrolyte interface (SEI) growth. However, the first cycle irreversible capacity loss yields low initial Coulombic efficiency (ICE) of Si, limiting the energy density. To address this, we adopt a prelithiation strategy to increase ICE and conductivity of all-solid-state Si cells. A significant increase in ICE is observed for $Li_1Si$ anode paired with a lithium cobalt oxide (LCO) cathode. Additionally, a comparison with lithium nickel manganese cobalt oxide (NCM) reveals that performance improvements with Si prelithiation is only applicable for full cells dominated by high anode irreversibility. With this prelithiation strategy, 15% improvement in capacity retention is achieved after 1000 cycles compared to a pure Si. With $Li_1Si$, a high areal capacity of up to 10 mAh $cm^{-2}$ is attained using a dry-processed LCO cathode film, suggesting that the prelithiation method may be suitable for high-loading next-generation all-solid-state batteries.

All-solid-state batteries (ASSBs) have drawn considerable attention as safer and potentially more energy-dense devices as compared to conventional liquid cells. Achieving high energy density ASSBs depends on the development of high-capacity electrodes in a solid-state architecture[1,2]. On the anode side, potential candidate materials or architectures include Li metal[3–6], anode-free[7], and alloy-type anodes such as Li-Si[8,9], Li-In[10–13], Li-Sn[14], Li-Al[15,16], Li-Sb[17], and Li-Mg[18]. However, high specific capacity and low propensity for Li dendrite growth and cell shorting make alloy-type anodes the most promising for next-generation ASSBs.

Si has been extensively studied in lithium-ion batteries (LIBs) for decades. Many reports have suggested that the use of pure Si as the anode is impractical due to its poor interfacial stability with liquid electrolytes and pulverization during cycling[19,20]. However, a recent study demonstrated the use of a 99.9 wt. % micro-silicon (µSi) anode in combination with an argyrodite solid electrolyte ($Li_6PS_5Cl$) to produce an ASSB with a high areal current density and high areal loadings[21]. The successful use of µSi as an anode was attributed to the passivation of the sulfide electrolyte-Si interface, limiting the growth of a poorly-conducting solid-electrolyte interphase (SEI).

Although Si-based all-solid-state cells with a passivating SEI and a high energy density have already been demonstrated, further performance improvements can be achieved, including increases in the initial Coulombic efficiency (ICE), electronic conductivity, and Li⁺ diffusivity (Fig. 1). Notably, the prelithiation of Si, which has traditionally been

[1]Materials Science and Engineering Program, University of California San Diego, La Jolla, CA 92093, USA. [2]Materials Department and Materials Research Laboratory, University of California, Santa Barbara, CA 93106, USA. [3]Insitute of Materials, Research, and Engineering, Agency of Science, Technology, and Research (A*STAR), Singapore, Singapore. [4]LG Energy Solution. Ltd., LG Science Park, Magokjungang 10-ro, Gangseo-gu, Seoul 07796, Korea. [5]Department of NanoEngineering, University of California San Diego, La Jolla, CA 92093, USA. [6]Department of Chemistry, Sogang University, Seoul 04107, Republic of Korea. [7]Pritzker School of Molecular Engineering, University of Chicago, Chicago, IL 60637, USA. ✉e-mail: jihyunjang@sogang.ac.kr; shirleymeng@uchicago.edu

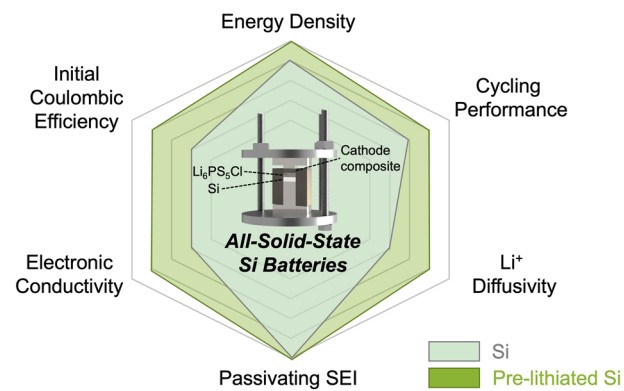

**Fig. 1 | Comparison of Si and prelithiated Si for all-solid-state batteries.** A radar comparison chart of Si (light blue shade) and prelithiated Si (green shade) anodes for various electrochemical properties and battery performance metrics.

implemented in LIBs[22], could be a good approach to enable such improvements.

One of the well-established prelithiation methods is electrochemical prelithiation[23–26]. In this approach, a Si electrode is lithiated by building an electrochemical cell comprising a Li metal counter electrode and a non-aqueous electrolyte. The redox potential difference of the two electrodes results in spontaneous lithiation of Si and SEI formation. However, the extent of "electrochemical" prelithiation must be well controlled, since insufficient lithiation cannot improve the ICE due to remaining Li trapping sites, while over-lithiation could start the lithium plating on the anode surface[27]. Due to the high chemical reactivity of Li, one of the challenges to implementing the successful prelithiation lies in finding stable prelithiation reagents (Li source). As such, Cao et al. introduced a polymer to protect the Li source, where the metallic Li source for prelithiation was shielded by the polymer before being made into the full cell[28]. More commercially viable option for prelithiation reagent is the stabilized lithium metal powder (SLMP). Ai et al. developed a solution process to coat SLMP on anode material, where both graphite/NMC and SiO/NMC full cells exhibited 31% higher ICE after the prelithiation of the anodes[29]. Forney et al. deposited SLMP on a Si – carbon nanotube (CNT) anode and used a mechanical press to apply a pressure of 100-300 PSI to the stack for 30-60 s to crack the electronically insulating $Li_2CO_3$ coating of SLMP and facilitate the prelithiation process[30]. Another study showed that mere contact between passivated Li metal powder (PLMP) and a Si/graphite electrode could induce prelithiation[31].

The first Si-based all-solid-state battery with prelithiation of the Si anode was recently reported[32], where prelithiation of Si was completed by ball-milling with Li metal in anhydrous hexane. Starting from the prelithiated Si, further mechanochemical milling was needed to mix the $Li_xSi$ active material, the solid electrolyte, and the carbon additive to form the composite anode. The $Li_xSi$ composite electrode was paired with a sulfur cathode and the full cell demonstrated a stable capacity for over 500 cycles. While such performance is impressive, this work required an extra high-energy ball-milling step in organic solvent to prelithiate the Si, adding complexity to the ASSB fabrication process. Prelithiation method using SLMP without electrolyte has been reported in the past as well. Jang et al. successfully implemented the prelithiation of Si without carbon paired with fluorinated polymer to mitigate SEI in liquid electrolyte[33]. Lee et al. enabled the SLMP-induced prelithiation of graphite-silicon without electrolyte in all-solid-state batteries[34].

Here, we introduce a simple pressure-induced prelithiation strategy for Si anodes during ASSB fabrication and the prelithiated Si anode was characterized using solid-state nuclear magnetic resonance (ssNMR). The performance of our prelithiated Si anode was evaluated in symmetric-, half-, and full-cells. In this work, the effectiveness of the prelithiation in ASSB was assessed depending on cathode selection and N/P ratio for the first time. Regarding long term cyclability, a cell of prelithiated Si paired with LCO showed a high ICE of over 95% with a stable cyclability for 1000 cycles at 5 mA cm⁻² current density.

Interestingly, we revealed that cathode irreversibility determined the effect of prelithiation on the full-cell and high N/P ratio Si cells behaved completely different from the liquid counterparts with the presence of excess Si. For solid-state cells, instead of having a low state of charge within the anode, Si becomes partially lithiated at its 2D interface and consistently acts like a cell with N/P ratio of 1. This behavior can be translated within a full cell, where the ICE was constant regardless of the N/P ratio. Moreover, the improved ICE was achieved even with a high loading of 10 mAh cm⁻² from the prelithiated Si, showing the true viability of the Si anode with a high-loading cathode. Based on the novel understanding, our work provides the insight to properly adopt prelithiated Si in ASSB configuration.

## Results and discussion

### Pressure-induced prelithiation of Si

Prelithiation of Si was conducted via a simple mixing process coupled with a pressurizing step. In this work, an anode composed of vortex-mixed μSi and SLMP was introduced for the first time in an ASSB. Different amounts of SLMP were mixed with μSi powder to produce $Li_xSi$ alloys with a molar ratio $x = 0.25, 1,$ and 2 (e.g., $Li_{0.25}Si, Li_1Si$ and $Li_2Si$). We note that those $x$ values assume that the SLMP in μSi powders have fully reacted. The morphology of μSi and SLMP was investigated with scanning electron microscopy (SEM), indicating a particle size distribution of 2–5 μm for μSi and 10–60 μm for spherical SLMP (Fig. S1a, b). From Fig. 2a, we find that Si and Li domains in the final $Li_1Si$ powder retain the morphology of the precursor particles. The absence of energy dispersive X-ray spectroscopy (EDS) signal from spherical regions within the $Li_1Si$ powder sample allow their assignment to pure lithium metal due to the low energy of the Li X-ray transition (Fig. 2a). In Fig. 2b, a cross-sectional focused ion beam scanning electron microscopy (FIB-SEM) image was obtained on a 200 MPa pressed $Li_1Si$ pellet. The pressed $Li_1Si$ sample exhibits two types of domains: 1) regions comprised of distinct μSi and Li sub-domains, and 2) regions where the μSi and Li precursors alloyed to form a new chemical composition. In the first type of domain, Li sub-domains are sandwiched between μSi domains, resulting in a different morphology from the SLMP precursor powder. Again, no EDS signal could be detected from those Li-rich sub-domains. The second type of domain has an entirely different morphology from the pristine μSi and SLMP powders, that is more comparable to charged (lithiated) Si where the gap between individual Si particle disappears and large Si blocks are formed instead. Additionally, Si EDS signal can be detected from those regions. Those analysis indicate that after pressing at 200 MPa for 30 s, the $Li_1Si$ anode exhibits unreacted Li and μSi, as well as a Li-Si alloy phase. During the pressure-induced lithiation, Si is lithiated by diffusion of Li which would follow the Fick's second law of diffusion at the contact point of Li and Si. However, the contact area of Li and Si particles is limited when the Li and Si mixture is pressed at low pressure (Fig. S2a). In Fig. S2b-c, the 400 MPa pressed $Li_1Si$ showed less remaining Li metal compared to 200 MPa the pressed pellet.

To better understand the extent of alloy formation from pressurizing SLMP and μSi precursors, ⁷Li ssNMR was used to probe the chemical state of bulk lithiated Si. ssNMR is crucial here, as lithiated Si is amorphous and cannot be studied using standard diffraction methods. ⁷Li ssNMR, on the other hand, is sensitive to crystalline and amorphous phases alike and allows to distinguish and, in theory, quantify Li metal (265 ppm)[35] from the Li-Si alloy (broad signal centered around 0 ppm)[31] as their respective signals are well resolved. However, the penetration of the radiofrequency (RF) pulses used to excite the nuclear spins in a ssNMR measurement into metallic samples

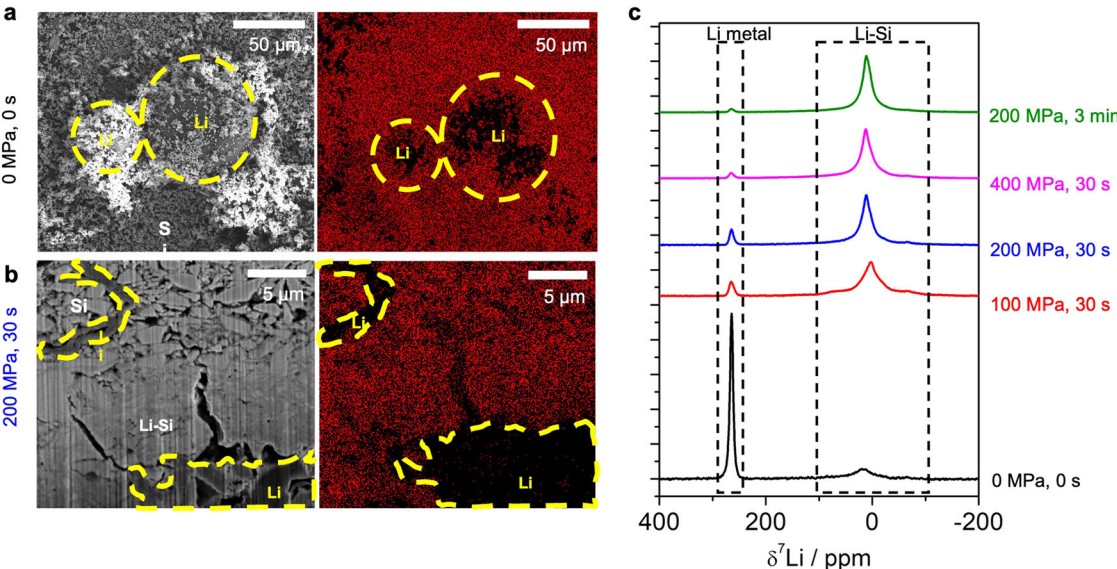

**Fig. 2 | Morphology and NMR spectra of pressure-induced lithiation of Si. a** FIB/SEM cross-sectional image of non-pressed (0 MPa, 0 s) and **b** pressed (200 MPa, 30 s) $Li_1Si$ pellet before cycling. **c** $^7Li$ NMR spectra of $Li_1Si$ with different pressure and time.

is limited and inversely proportional to the square root of the electronic conductivity. This results in a so-called "skin depth" of around 7.4 µm for Li metal (with an electronic conductivity of $1.1 \times 10^7 \text{S/cm}$)[36] under the chosen experimental conditions[37], which is lower than the radius of some pristine SLMP particles (5–30 µm) and thus leads to an underestimation of the amount of metallic Li in the sample. The Li-Si alloy phases that form in the samples under consideration are many orders of magnitude less conductive than Li metal, as will be discussed in the next section, and their minimum skin depth of ~570 µm (calculated based on $Li_2Si$ electronic conductivity) ensures that such regions can be probed quantitatively with $^7Li$ ssNMR. We note, however, that the Li-Si alloy signal likely overlaps with minor diamagnetic impurity phases that inevitably form at the surface of metallic Li (even when air/moisture exposure was avoided by handling the samples in the glovebox at all times), including LiOH and $Li_2CO_3$ resonating at ~0 ppm[38]. In Fig. 2c, $^7Li$ ssNMR was conducted on several SLMP + µSi samples subjected to pressures varying from 0 to 400 MPa for 30 s to 3 minutes to determine the conditions under which maximum Si pre-lithiation is achieved. Given that the size distribution of metallic Li subdomains in the pressed samples is on par with the particle size distribution of the SLMP precursor, the relative amount of Li metal detected by $^7Li$ ssNMR follows the same trend as the actual amount of metallic Li in the samples of interest to this work, despite skin depth issues. This allows us to use the relative integrated intensity of the Li metal and Li-Si alloy signals in the $^7Li$ ssNMR spectra to follow the incorporation of Li into Si as a function of sample processing conditions. The impact of ssNMR signal ($T_2^*$) relaxation during data acquisition was accounted for in the quantification of the observable Li signals (see results in Table S1) for all samples except the unpressed $Li_1Si$ for which a $T_2^*$ measurement could not be conducted due to sample evolution in the spectrometer. The unpressed $Li_1Si$ sample contains the most Li metal and a smaller fraction of the Li-Si phase, with the 0 ppm resonance accounting for 27.8 % of the total $^7Li$ ssNMR signal intensity. Despite the lack of $T_2^*$ adjustment for this sample, these results are expected to hold as the changes in Li molar % from $T_2^*$ adjustment of Li metal or diamagnetic phases for other samples are smaller than 1%. Conversely, pressed $Li_1Si$ samples exhibit > 92% of the total $^7Li$ signal intensity at 0 ppm, indicating the presence of a major Li-Si alloy phase. The relative intensity of the 0 ppm signal as compared to the Li metal signal increases with higher pressure and longer pressing time, indicating an increased fraction of Li-Si alloy in the

sample. For example, applying 200 MPa of pressure for 3 min (green) leads to greater Li incorporation into the Si phase than applying the same amount of pressure for 30 s (blue), as evidenced by the 98.2 and 92.9% of the total $^7Li$ ssNMR signal intensity present at 0 ppm for these two samples, respectively. Those results indicate that a Li-Si alloy can be formed by pressurizing the SLMP and µSi precursor powders in the absence of electrolyte, demonstrating that the latter is not required to facilitate the alloying reaction unlike previously thought[39]. Interestingly, unpressed $Li_xSi$ samples evolve over the course of the ssNMR measurements, while the composition of pressed samples remains stable. This is shown in Fig. S3, where $^7Li$ ssNMR spectra collected on four different $Li_xSi$ samples ($Li_1Si$ and $Li_2Si$ non-pressurized and pressurized at 200 MPa for 30 s) before, during, and after a $T_2^*$ relaxation time measurement, are compared. For the unpressed samples, the 0 ppm signal increases over the course of the measurement, indicating that Li-Si alloying is taking place over time. On the other hand, the spectra of the pressed samples do not evolve because Si has already been lithiated at 200 MPa and is stable under ambient conditions. The homogeneity of NMR spectra with the fitting is provided in Fig. S4.

## Electrochemical comparison of $Li_xSi$ in symmetric, half, and full cells

Although Si is a semiconductor, its low electronic conductivity (in the range of $10^{-4} \text{ S cm}^{-1}$), results in a large overpotential within the cell. The conventional way to overcome this barrier is to add carbon or some conductive agent, creating a silicon composite anode. However, the addition of Li into silicon could be another way to increase the electronic conductivity of Si dramatically. Figure 3a shows that as Li content increases, the $Li_xSi$ electronic conductivity increases from $10^{-4}$ ($Li_0Si$) to $10 \text{ S cm}^{-1}$ ($Li_2Si$). Since pressure-induced lithiation of Si was proven to be an effective approach from the previous section, we evaluated the electrochemical properties of $Li_xSi$ in the cell configuration of symmetric, half and full cells. In Fig. 3b, the plating and stripping of $Li_xSi$ symmetric cells were conducted to evaluate the overpotential of each cell. The overpotentials of cells decrease with more Li in Si, which is consistent with higher electronic conductivity of higher Li content Si from Fig. 3a. The high electronically conductive $Li_2Si$ symmetric cell had much smaller overpotential than $Li_{0.25}Si$ symmetric cell. Electrochemical impedance spectroscopy (EIS) was used to evaluate the resistance of $Li_xSi$ symmetric cells before and after lithiation and

delithiation respectively in Fig. 3c, d. Before plating and stripping, the resistance of $Li_{0.25}Si$ is higher than that of $Li_1Si$ or $Li_2Si$ due to the poor contact between two electrodes and LPSCl electrolyte pellet because of stiffness of low lithiated silicon (please note that the resistance of around 35 Ohm mainly comes from the ionic conductivity of LPSCl pellet between two electrode ( ~ 2.2 mS $cm^{-1}$). The resistance value slightly decreased after plating and stripping, maintaining the trend of higher Li content in Si resulting in lower impedance. The resistance values of $Li_1Si$ and $Li_2Si$ before and after plating were comparable to the ionic conductivity of the sulfide electrolyte $Li_6PS_5Cl$ (LPSCl) electrolyte layer, indicating $Li_1Si$ nor $Li_2Si$ is not a dominant component of the cell resistance. Figure 3e shows the half-cell configuration of $Li_xSi$ with Li metal. All $Li_xSi$ was first lithiated for 1-hour and then delithiated for 1-hour at the same current density. Interestingly, all $Li_xSi$ exhibited similar overpotential during the lithiation, indicating that Li reacting with Si into $Li_xSi$ requires a similar amount of overpotential. The non-

prelithiated Si clearly showed a higher overpotential when it was first lithiated due to its poorer electronic conductivity. However, the overpotential of the delithiated process is prominently different depending on the degree of prelithiation. This indicates that the amount of prelithiation eventually affects the electronic/ionic conductivity of silicon during charge and discharge. There are two sources of irreversible capacity during the first cycle; one is the electrolyte decomposition on the interface and the other is Li trapped inside Si[21], which are successfully compensated for by our prelithiation strategy.

In Fig. 4a, full cells with the following configuration, $Li_xSi$ | LPSCl | LCO, were fabricated and cycled at C/20 to study the effect of various prelithiation amounts in Si, which was to evaluate the first cycle performance with limited lithium inventory. Although the charge capacities of all $Li_xSi$ were similar, the discharge capacity of $Li_xSi$ showed significant differences. This result is also reflected in the half-cell configuration in Fig. 3e, where lithiation of Si (charging) is comparable

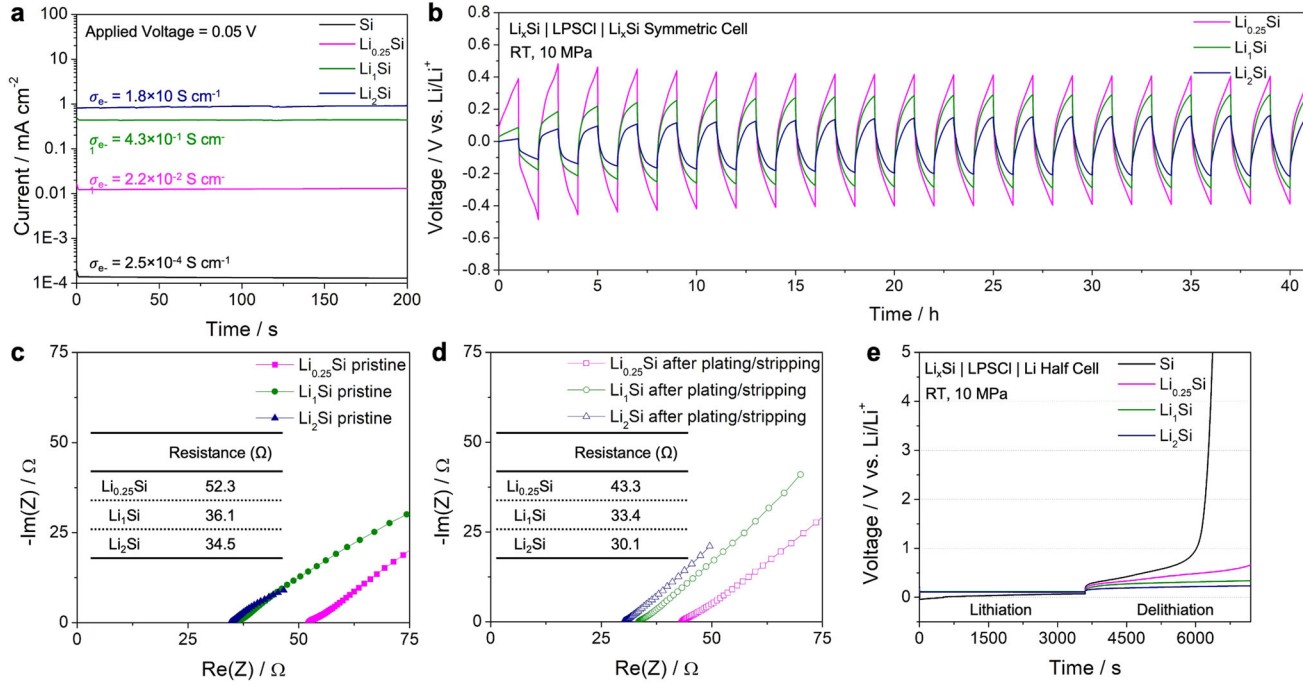

**Fig. 3 | Conductivites and electrochemical properties of $Li_xSi$ in symmetric and half-cells. a** The electronic conductivity of Si and vortex mixed $Li_1Si$ using direct current polarization. **b** Plating and stripping of $Li_xSi$ (x = 0.25, 1, and 2) for 20 cycles at 0.2 mA $cm^{-2}$. **c** EIS measurement of $Li_xSi$ symmetric cell before plating/stripping. **d** EIS measurement of $Li_1Si$ symmetric cell after plating/stripping at 0.2 mA $cm^{-2}$. **e** Lithiation and delithiation of $Li_xSi$ half-cells with different lithiation states.

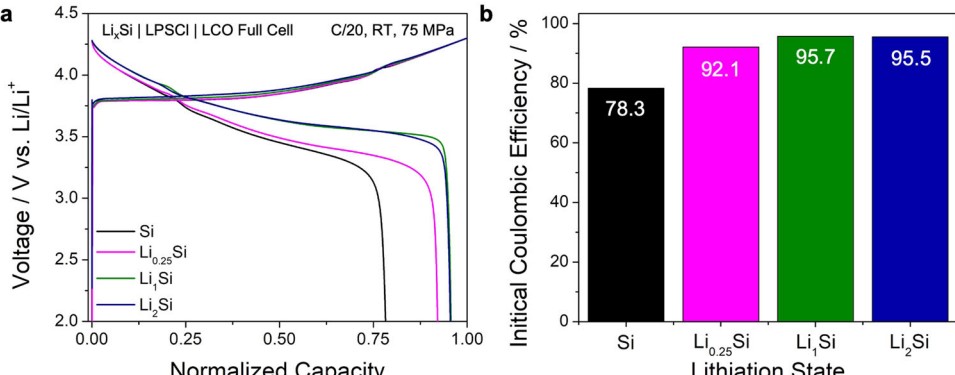

**Fig. 4 | First cycle performances of $Li_xSi$ full-cells. a** 1st cycle voltage curve of $Li_xSi$ full cells with different lithiation states. LCO | LPSCl | $Li_xSi$ cells were cycled at room temperature and 75 MPa. **b** Initial Coulombic efficiency trend of $Li_xSi$ (x = 0, 0.25, 1, and 2).

but delithiation of Si (discharging) shows a dramatic change in ICE. The ICE of the LCO-Si full cell was 78.3% whereas the ICE of LCO-Li$_1$Si and LCO-Li$_2$Si was increased to over 95% (Fig. 4b).

## Cathode limiting or anode limiting: NCM811/LCO and Si/Li$_1$Si

Two different cathodes were paired with Si and Li$_1$Si to elucidate the limiting component of the system (Fig. 5). Based on the half-cell data of each component with a Li counter electrode, we can assume ICE of **NCM** as 75%, Si as 80%, and LCO as 95% (Fig. S5). For NCM and LCO cathode paired with Si and lithiated Si, we can assume four cases. (Fig. 5a) For the NCM/Si full-cell, the overall ICE is limited by the ICE of NCM, while the Si ICE determines the ICE of the LCO/Si full-cell. Therefore, pairing NCM with lithiated Si with excess Li on the anode side, the cell will still be limited by the ICE of NCM and will be unable to utilize the excess Li. However, by pairing LCO with lithiated Si, we can utilize the excess Li during the first discharge, and the cell can reach the ICE limit of LCO yielding 95%. Therefore, Case 1 (NCM/Si) and 3 (NCM/Li$_x$Si) can be regarded as the cathode-limiting system, while Case 2 (LCO/Si) is anode-limiting system. This means that prelithiation is only effective if the full-cell is anode limited. Cells corresponding to

each of these four cases were fabricated to demonstrate this hypothesis. In Fig. 5b, which shows the NCM811 case, the ICE improvement at C/20 was marginal after prelithiation. However, the ICE of LCO cells increased significantly from 78.3% to 95.7% (Fig. 5c). The first-cycle voltage profiles from these cells were consistent with the hypothesis illustrated in Fig. 5a Case 2 and Case 4. As Fig. 5a Case 4 achieved the highest ICE of 95.7%, the further long cycling and higher loading efforts are all made in this configuration. From the rate tests in Fig. S6a and b, lithiated Si always showed higher discharge capacity than non-lithiated Si for all current densities.

There is one more important point regarding the N/P ratio. Although the illustration in Fig. 5a explained the ICE of full-cell depending on the cathode-/anode-limiting system based on the N/P ratio of around 1, we obtained the experimental results (Fig. 5b, c) at the relatively high N/P ratio of 4.4. A high N/P ratio generally decreases ICE since irreversible lithium/electron consumption happens at a relatively high voltage (the initial stage of the lithiation process). However, our results show that the full-cell which has a wide range of N/P ratio (1–3.3) exhibits similar ICE values (Fig. S7), because some anode parts practically don't participate in the lithiation process

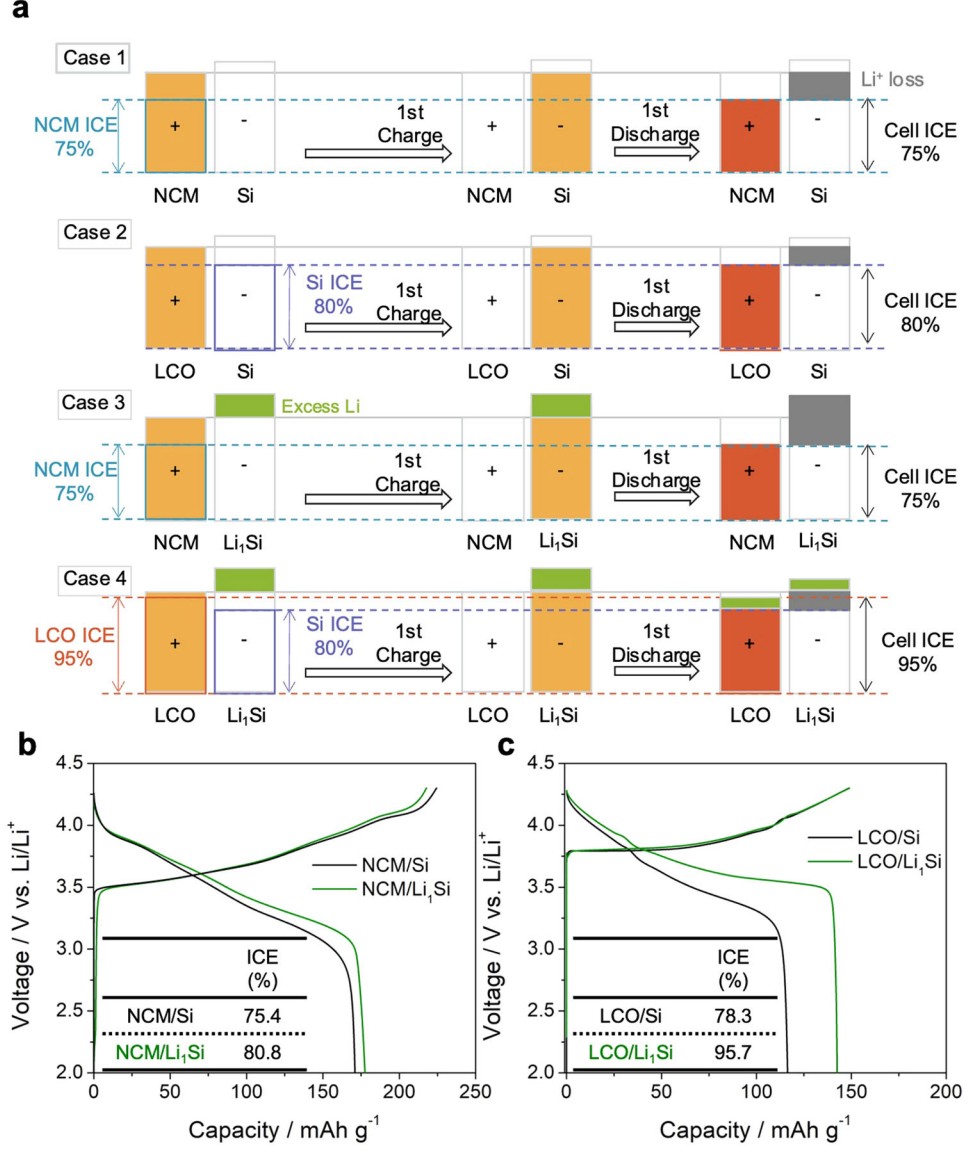

**Fig. 5 | Prelithiation driven improvement of cathode anode limiting cases. a** Schematic illustrating ICE estimates of the Si and Li$_1$Si paired with NCM and LCO cathodes. First-cycle voltage profiles of (**b**) NCM811 and (**c**) LCO paired with Si and Li$_1$Si at C/20.

(Fig. S8). It makes the effective N/P ratio of our solid-state cell around 1, consistent with the illustration.Therefore, the discussion at the beginning of this section of Case 1 to Case 4 is valid even with higher loading of Si.

The morphology of $Li_1Si$ upon charging and discharging was demonstrated in Fig. S9. The charged $Li_1Si$ cross-sectional image shows that the partial utilization of Si is valid even in the $Li_1Si$ case, meaning only the LPSCl facing side of $Li_1Si$ gets lithiated whereas the opposite current collector side still has unreacted Li metal (dark) as we discussed earlier. The discharged sample exhibited surface cracks in some parts of the electrode (Fig. S9f) where the cross-sectional image of the non-cracked part was shown in Fig. S9d and the cracked part shown in Fig. S9e, which indicated the existence of a huge volume change of the silicon electrode. In addition, the volume expansion from pristine to charged state was ~200% (Fig. S9), and the discharged state showed minimal difference in thickness compared to the pristine state. The volume expansion rate seems to be below the reported lithiated Si, but this is mostly because $Li_1Si$ was partially lithiated where part of the anode was not utilized. Still, the relatively lower volume expansion rate could benefit the long-term cycling of $Li_1Si$.

## Ramping test and long cycling of prelithiated Si

A ramping test using Si (Fig. 6a) and $Li_1Si$ (Fig. 6b) was conducted to evaluate the lithiation effect on critical current density. Both Si and $Li_1Si$ did not short, even up to $10 \, mA \, cm^{-2}$. The areal capacities of a LCO cathode composite in all cells were $4 \, mAh \, cm^{-2}$. For the first cycle at $0.25 \, mA \, cm^{-2}$, both cells showed similar charge capacity suggesting good utilization of the cathode materials from the same loading. However, from the first discharge step, the difference in capacity begins to dominate, which is always higher when paired with $Li_1Si$. The cycling stability of both Si and $Li_1Si$ full cells, cycled at $5 \, mA \, cm^{-2}$ is shown in Fig. 6c. Even with the high rate of 1.25 C (1 C = $4.0 \, mAh \, cm^{-2}$), the retention of $Li_1Si$ cell was 73.8% after 1000 cycles with an average CE of 99.9%, whereas the non-lithiated Si cell demonstrated 58.7%

retention after 1000 cycles. As discussed in Fig. 3e, for the Si full-cell, decrease in reversible capacity originates from electrolyte decomposition at the interface (especially LPSCl/Si interface) and Li trapped in Si. Even though stabilized LPSCl/Si interface after first few cycles helps the Si full-cells to have excellent CE and cyclability, it can be clearly seen that cells with more Li inventory (excess Li by prelithiation) have better cyclability. This further supports the room temperature lithiated Si could work at high rates for extended cycling.

Interestingly, the cycling trend of lithiated Si shows an initial discharge capacity increase rather than decay. To better understand the full cells, EIS was measured for both cases upon cycling. This initial increase in discharge capacity could be attributed to residual Li metal not lithiated to $Li_xSi$ which then becomes lithiated electrochemically in subsequent cycles. In Fig. S10, in-situ EIS of the full cell using both Si and $Li_xSi$ shows a decrease in resistance as it cycles. However, the magnitude of the resistance decrease is much higher for the $Li_xSi$. This implies that the remaining Li in $Li_xSi$ would keep lithiating the unlithiated Si as it cycles.

Considering the amount of Si used in the cell, all cells exhibit a high N/P ratio. The amount of Si used was fixed to 5 mg for all cells, yielding ~14 mAh of theoretical anode capacity and N/P ~ 4.4. As such, increasing the cathode loading to match the high capacity of anode was needed. However, with regards to the high-loading thick electrode, an inhomogeneous reaction within the thick electrode has been reported previously, showing lithium-ion diffusion limitation which resulted in the state of charge variation[40,41]. Therefore, a dry processing of cathode film consisting of cathode, catholyte and polytetrafluoroethylene (PTFE) binder was fabricated to achieve a better homogeneous electrochemical pathway within the thick electrode[42]. The cathode loadings were further increased in Fig. S11, from dry process LCO loading of 22 mg to 42 mg and 57 mg, each corresponding to $3.7 \, mAh \, cm^{-2}$, $8.0 \, mAh \, cm^{-2}$ and $10.8 \, mAh \, cm^{-2}$ of theoretical cathode capacity. The areal capacity from three different cathode loadings corresponds well with these theoretical cathode capacities

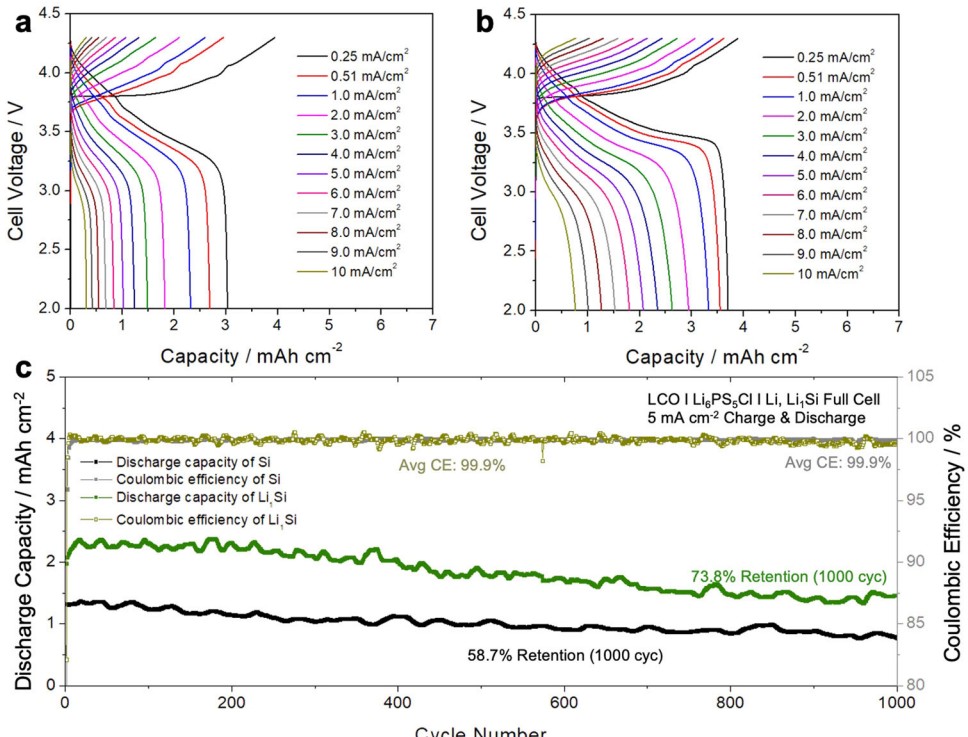

**Fig. 6 | Cycling performance of Si and $Li_1Si$ full cells.** Ramping test to evaluate the critical current density of (**a**) Si and (**b**) $Li_1Si$. **c** Cycling performance of Si and $Li_1Si$ cell at $5 \, mA \, cm^{-2}$.

(Fig. S11a). The gravimetric capacity of two higher loading cells decreased by 10 mAh g$^{-1}$, however, even for the 10 mAh cm$^{-2}$ cell, capacity close to theoretical capacity of LCO was achieved. This demonstrates the high capacity of lithiated Si in the full cell configuration. The energy density of the high loading cell has been calculated to be 236 Wh kg$^{-1}$ and 947 Wh L$^{-1}$ for the high loading 10.8 mAh cm$^{-2}$ cell shown in Fig. S11, which is based on the assumption that the solid electrolyte layer is 30 μm.

All-solid-state Si batteries have shown promising potential to enable high-capacity anode without continual SEI growth. However, the low ICE of Si remained a challenge to overcome for all-solid-state batteries. Here, a prelithiation strategy using the stabilized lithium was adopted to improve the ICE and conductivity of anodes. The lithiated Si was examined in symmetric-, half-, and full-cell configuration to understand the cell- level improvement of each component. With Li$_1$Si and LCO used as the anode and cathode, respectively, the full cell showed over 95% of ICE. In this work, we have identified why the prelithiation effect would dominate only for the anode-limited cases by comparing NCM and LCO paired with Si and Li$_x$Si. The ramping test and the long cycling performance were evaluated for both Si and Li$_1$Si cells. The Li$_1$Si demonstrated a large improvement of 73.8% after 1000 cycles, a 15% improvement in retention. Furthermore, using Li$_1$Si, a high areal capacity of 10 mAh cm$^{-2}$ was achieved and demonstrated using a dry-process LCO film, demonstrating that the lithiated Si could be a suitable candidate to be used in high-energy-density next-generation batteries.

## Methods

### Materials preparation

Li$_6$PS$_5$Cl (LPSCl, NEI Corporation, USA) was used for the solid-state electrolyte (SSE) separator layer and cathode composite preparation. For cathode composite purposes, the LPSCl particle size was reduced using an E$_{MAX}$ ball mill (Retsch, Germany). The ball milling was conducted for 2 hours at 300 rpm, using anhydrous xylene as a medium. Lithium cobalt oxide (LCO, MSE Supplies, USA), coated with a niobium-based layer, was used as received. The cathode composite was prepared by hand-mixing using a weight ratio of LCO: LPSCl = 70: 30.

For the preparation of lithiated μSi, μSi (Sigma Aldrich, USA) and stabilized Li metal powder (FMC, USA) was vortex mixed for 3 min. The mixture was subsequently pressed using a hydraulic press at 100 MPa for 30 s.

### Materials characterization

For the $^7$Li solid-state NMR (ssNMR) measurements, all one-dimensional spectra were acquired at 18.8 T (800 MHz for $^1$H) on a Bruker Ultrashield Plus standard bore magnet equipped with an Avance III console. The measurements were carried out using a 3.2 mm HXY MAS probe, and 3.2 mm single cap zirconia rotors packed and closed with a Vespel cap under Ar with a PTFE spacer between the sample and cap to further protect the sample from air exposure. A flow of N$_2$ gas at 2000 L h$^{-1}$ was used to protect the sample from moisture contamination. Data were obtained using a static spin-echo pulse sequence (30°-TR-60°-TR-ACQ) with a 10 μs echo delay (TR). Rotors were kept static throughout each measurement to avoid sample evolution caused by frictional heating during magic angle spinning (MAS). 30° and 60° flip angles of 1.617 μs and 3.234 μs at 200 W, respectively, were used, and recycle delays of 10-90 s between scans were applied according to the longitudinal (T$_1$) relaxation properties of the sample. $^7$Li chemical shifts were referenced to a 1 mol/L LiCl liquid solution at 0 ppm. Pulse lengths were calibrated on a liquid solution consisting of 80% volume saturated LiCl and 20% volume 1 mol/L CuSO$_4$. All spectra were processed with Topspin 3.6 and fitted with an in-house python code. T$_2$* measurements on each sample were also conducted to compensate for uneven signal decay of the Li metal and diamagnetic components during the 10 μs echo delay. On each sample, a series of static spin-echos (30°-TR-60°-TR-ACQ) with variable echo delays was acquired and the spectra were integrated from 240-280 ppm and –200-200 ppm to account for Li metal and the overlapping diamagnetic signals, respectively.

The cross-sectional images were obtained using the FEI Scios Dualbeam (ThermoFisher Scientific). To ensure minimal air exposure during the sample transfer, an air-tight transfer arm was employed to move the sample from the Ar-filled glovebox into the FEI Scios Dualbeam chamber. Subsequently, to mitigate the beam damage, liquid nitrogen and a heat exchanger were utilized to maintain the cryogenic temperatures during the ion beam milling and electron beam imaging. The focused ion-beam of gallium ions was used to mill the cross-sections of samples, with the milling parameters set to 30 kV and 65 nA, and the cleaning parameters adjusted to lower currents of 30 nA and 15 nA. The settings for imaging using electron beam were chosen to be 5 kV and 0.1 nA.

### Electrochemical characterization

Two titanium rods were used as current collectors at each end of the Li metal. The solid-state separator layer was fabricated by first putting 75 mg of LPSCl in a 10 mm inner diameter polyether ether ketone holder, which was then compressed between two titanium rods at 370 MPa. LCO cathode composite of 30 mg (active loading of 26.7 mg/cm$^2$) was placed on top of the LPSCl separator pellet and pressed at 370 MPa using a hydraulic press. 5 mg of Si was put for all Si and lithiated Si cells in this work. Si and Li$_x$Si were inserted onto the other side of the LPSCl separator pellet and pressed at 100 MPa. The cells were set to 75 MPa before cycling started. The full cell configuration follows the same protocol except one Li side is replaced with a cathode composite. All cell cycling was performed at room temperature using in the Argon-filled atmosphere glovebox. The battery cells were cycled using a Neware Battery cycler and analyzed with BTS900 software. EIS measurements were conducted using Biologic SP-200. The frequency range was from 10 MHz to 0.1 Hz, with an applied AC potential of 10 mV. Direct current polarization was conducted to measure the electronic conductivity of Li$_x$Si by applying the voltage of 100 mV for 3 min.

## Data availability

All data supporting the findings of this article and its Supplementary Information will be made available upon request to the authors.

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

## Acknowledgements

This work was funded by the LG Energy Solution through Frontier Research Laboratory (FRL) program. This work was performed in part at the San Diego Nanotechnology Infrastructure (SDNI) of UCSD, a member of the National Nanotechnology Coordinated Infrastructure, which is supported by the National Science Foundation (Grant ECCS-2025752). The research reported here made use of the shared facilities of the Materials Research Science and Engineering Center (MRSEC) at UC Santa Barbara: NSF DMR-2308708. The UC Santa Barbara MRSEC is a member of the Materials Research Facilities Network (www.mrfn.com). E.S. acknowledges support from the National Science Foundation Graduate Research Fellowship under Grant NSF DGE 1650114.

## Author contributions

So-Yeon Ham contributed concenptualization, data curation, methodology, formal analysis, investigation, writing – original draft, visualization. Elias Sebti contributed data curation, formal analysis, investigation, writing. Ashley Cronk contributed formal analysis, investigation. Tyler Pennebaker contributed data curation, investigation. Grayson Deysher contributed: formal analysis, validation. Yu-Ting Chen contributed investigation, visualization. Jin An Sam Oh contributed investigation, validation. Jeong Beom Lee contributed formal analysis, investigation.

Min Sang Song contributed formal analysis, investigation. Phillip Ridley contributed formal analysis, investigation. Darren H. S. Tan contributed conceptualization, methodology, investigation. Raphaële J. Clément contributed formal analysis, investigation, writing. Jihyun Jang contributed conceptualization, formal analysis, writing. Ying Shirley Meng contributed conceptualization, investigation, writing.

## Competing interests

The authors declare no competing interests.
