## [Peer Review File · Nature Communications]

Overcoming Low Initial Coulombic Efficiencies of Si Anodes Through Prelithiation in All-solid-state BatteriesREVIEWER COMMENTS

Reviewer #1 (Remarks to the Author):

The paper reports high initial efficiency (ICE) of silicon (Si) anode in all-solid-state batteries using prelithiation method. While it is true that the all-solid-state full-cell and half-cell initial efficiencies were significantly increased, the authors use the well-known SLMP mixing method and the experiments only confirm what is already known, such as ICE and conductivity improvement through lithiation of silicon; I can't find significant scientific novelty. Due to the lack of novelty and details in the logical arguments, I would not support the publication of this work in Nature Communication.

1. The authors claim that mixing SLMP and Si for prelithiation using vortex mixing is a novel approach. However, it's not clear why this method is significant, as it involves additional steps such as high-pressure pressing after mixing. Moreover, prelithiation of active materials using SLMP without electrolyte has been reported several times previously [Journal of Power Sources 580 (2023): 233326. / Advanced Energy Materials (2023): 2300172.]. Therefore, the scientific novelty of the prelithiation method used in this study is questionable and requires further consideration.

2. In Figure 3, the authors suggest that the degree of prelithiation results in differences in conductivity, overpotential, and resistance. However, these differences based on Li-Si composition are already well-documented several times [Journal of Power Sources 580 (2023): 233326. / Nature Energy (2023): 1-14. / The Journal of Physical Chemistry C 115.5 (2011): 2514-2521.]. This experiment does not contribute new insights to the existing literature. Additionally, the overpotential of symmetric cell (Figure 3b) and half-cell (Figure 3e) is also influenced by the voltage of the Si active material depending on the state of charge (SOC), which is not addressed in the study. Further investigation into this phenomenon is needed.

3. In Figure 5, it is claimed that the ICE enhancement of full-cell by prelithiation occurs when anode-limiting. This is a reasonably known fact, but it is quite contradictory that the study uses a very high n/p ratio of 4.4, and yet the efficiency of the NCM full-cell is dominated by the cathode. Since Figure 5 only considers the case of np ratio of 1, further explanation is needed to consider the np ratio used in the actual cell.

4. In figure 6, long cycle life of full cell and efficiency data fluctuates a lot. This can also lead to reliability issues in cell evaluation.

Reviewer #2 (Remarks to the Author):

The authors reported using Li powder to prelithiate micron size Si, which was subsequently used in a solid-state battery. Although using powder Li to prelithiate cathode e.g. hard carbon or even si is nothing new, using powder Li to lithiate Si for a solid-state battery is new. The selling point of the manuscript is using pressure to gain better prelithiation, which is no brainer. Having those said, I still think it can be published if the author can conduct more investigation to reveal detail mechanisms.

1) The kinetics of Li diffusion under different pressure.

2) The change of Li concentration gradient with a Si particle with time (Fick's second law may be used).

3) Since the stabilized Li powder was used, how the residue from the Li particle impacted the Si conductivity. 45 ohm impedance in a symmetric cell seemed high (figure 3 (c)). The charge and discharge curves (Figure 3 b) also seemed strange.

4) It is strange that the authors use either LCO and NMC811. Please provide the rational regarding how the full cell was balanced.

5) Please provide more detail analysis on impedance data, e.g. whether the data is in compliance with linearity? Separation of different electrochemical processes.

Reviewer #3 (Remarks to the Author):

Please refer to the attached file.

Reviewer #4 (Remarks to the Author):

The manuscript reports the preparation of a lithiated Si electrode and its investigation in various cell configurations using Li-argyrodite as electrolyte material. The authors demonstrate that in ASSB the use of prelithiated Si electrodes improves the cycling performance, especially if the capacity of the anode was limiting. Impressive cycling performances as well as higher CAM loading through dry processing techniques are presented by the authors. This manuscript can be accepted after addressing the following issues.

1. In Figure 1 presents the spider diagram of the characteristic properties of Si over prelithiated Si for ASSBs. The color choice makes it difficult to differentiate between the Si and prelithiated Si electrodes. The Initial CE is linked with the formation of the passivation layer on Si particles. Why in the case of Si and prelithiated Si are no differences observed? Furthermore, using Si and prelithiated Si just a small increase is observed for the energy density and in the manuscript text only ICE, electronic conductivity and Li diffusion are mentioned. Can you please comment why energy density is not mentioned or discussed, as based on the material characteristic properties some hypothetical calculations can be made?

2. In the introduction part, the focus is made on the pre-lithiation method. However, in the case of Si and solid-state electrolytes the discussion is not mentioning with too many details the effect of pre-lithiation on the material polarizability. Can you please comment if the prelithiation, in this case, is made exclusively in order to improve the Li conductivity in the high Si-loading anode material?

3. Figure 2b and Figure 2c indicate the evolution of the electrode morphology and the change of Li metal environment from SLMP to prelithiated Li-Si phase. Can you comment on the homogeneity of the assigned-Li₁Si phase throughout the electrode bulk after applying pressure from the ss-NMR perspective? Are there any resonance shifts of the Li-Si regions that can be assigned to other Li-rich-Si phases that are formed during the 30s or 3 min applied pressure, and later evolving into Li₁Si phase?

4. At page 5, the authors mention that the Li-Si alloy phases can be quantified from the Knight shift around 0ppm. Is the 0ppm shift not characteristic to Li in the SEI while Li-Si phases, according to Key et al., J. Am. Chem. Soc. 2009, 131, 26, 9239–9249, should be around 18- to 6ppm? Can you comment?

5. In Figure 3a, the addition of 0.25 equivalents of Li already increases the electric conductivity to almost two orders of magnitude compared to pure Si electrode. The effect is also observable in Figure 3e, where it seems that the overpotential decreases to negative values (Li plating?) before some uptake of Li is occurring. Can you please comment?

6. At page 7, the following statement is made: "This indicates that it is harder to delithiate Li_xSi depending on how much Li is in Si.". How this statement aligns with the behavior of Si|LPSCI|Li cells?

7. In the next sentence : "In addition, the irreversible capacity from the initial lithiation and delithiation was observed in Li_xSi paired with Li counter electrode." the discussion is made upon the ICE or CE of the Li_xSi paired with Li. The value of the ICE or CE of the Li₁Si|LPSCI|Li should be provided in the text (also for the sake of the discussion in subsection 2.3).

8. In Figure 4 only the initial cycle is presented. This should be clearly indicated also in the manuscript text. The Axis label ICE (Figure 4b) should be corrected to "Initial".

9. The title of the subsection 2.3 should report the N/P ratio. The authors use NCM811/LCO or Si/Li₁Si that is confusing. What is the areal capacity of the Si or Li₁Si electrodes?

10. In subsection 2.3, the notation Li_xSi should be referred to the composition of the negative electrode, i.e. Li₁Si (according to the results, only the phase Li₁Si and Si electrodes were used). In subsection 2.2 it is appropriate to use Li_xSi.

11. Figure 3e and Figure S3 show the half-cell data of Si|LPSCI|Li cells. Why the evolution of the voltage profile is different? Also, in the caption 10MPa are mentioned to be used for the applied

pressure while 15MPa are indicated in the Figure S3. Please correct.

12. The case 4 in Figure 5a is not discussed in the text. Since this is the outcome of the manuscript is based on this effect, the authors should further support the schematic with significant ICE values and discussion in the subsection 2.3.

13. In Figure 6c, the discharge capacity decay seems to be steeper for the Li₁Si cell, rather than for Si. The authors explain the increase during the initial <20 cycles for the Li₁Si cell as a result of the residual Li present in these electrodes. Why in the cycling stability plots of both materials, such increase and decrease occur for both samples?

14. Are the values for the current density used in Figure 3e and Figure 6c similar?

15. In subsection 2.3 the authors concluded that the prelithiation is only effective if a limiting anode is used. How is this statement supportive to the last paragraph of the manuscript?

16. In the last part of the manuscript, the authors present the effect of higher active material loading for the cathode material. It is remarkable that the authors achieve such high loading. The authors should present and briefly discuss the effect of such high mass loadings on the kinetics of the cathode material as well as on the long cycling performance.

REVIEWER COMMENTS

Reviewer #1 (Remarks to the Author):

The paper reports high initial efficiency (ICE) of silicon (Si) anode in all-solid-state batteries using prelithiation method. While it is true that the all-solid-state full-cell and half-cell initial efficiencies were significantly increased, the authors use the well-known SLMP mixing method and the experiments only confirm what is already known, such as ICE and conductivity improvement through lithiation of silicon; I can't find significant scientific novelty. Due to the lack of novelty and details in the logical arguments, I would not support the publication of this work in Nature Communication.

We appreciate the reviewer's constructive feedback. The comments have been carefully considered and are addressed individually in the responses below.

1. The authors claim that mixing SLMP and Si for prelithiation using vortex mixing is a novel approach. However, it's not clear why this method is significant, as it involves additional steps such as high-pressure pressing after mixing. Moreover, prelithiation of active materials using SLMP without electrolyte has been reported several times previously [Journal of Power Sources 580 (2023): 233326. / Advanced Energy Materials (2023): 2300172.]. Therefore, the scientific novelty of the prelithiation method used in this study is questionable and requires further consideration.

The reviewer has raised an insightful point. As the reviewer pointed out, prelithiation using SLMP without electrolytes has been reported. However, we believe that our work still has the novelty as follows:

- i. The prelithiation step does not require additional steps during cell fabrication. The pressurizing of solid-state pellets simultaneously prelithiate the Si anode. The pressure and time required for the prelithiating Si during the fabrication were quantitatively investigated using NMR.
- ii. The work achieved a high ICE of 95% with a high loading of 10 mAh cm⁻² in the all-solid-state cell using prelithiated Si.
- iii. The added section of N/P ratio consideration highlighted the main difference between solid-state and liquid cells in high N/P (Si excess).
- iv. There is no paper yet to realize all-solid-state Si without carbon and electrolyte achieving highly improved ICE.

The papers mentioned by the reviewers are cited and discussed in the main text as below.

“Prelithiation method using SLMP without electrolyte has been reported in the past as well. Jang et al. successfully implemented the prelithiation of Si without carbon paired with fluorinated polymer to mitigate SEI in liquid electrolytes.³³ Lee et al. enabled the SLMP-induced prelithiation of graphite-silicon without electrolyte in all-solid-state batteries.³⁴”

33. Jang, E., Ryu, S., Kim, M., Choi, J. & Yoo, J. Silicon-stabilized lithium metal powder (SLMP) composite anodes for fast charging by in-situ prelithiation. *Journal of Power Sources* **580**, 233326 (2023).

34. Lee, J. et al. Dry Pre-Lithiation for Graphite-Silicon Diffusion-Dependent Electrode for All-Solid-State Battery. *Advanced Energy Materials* **13**, 2300172 (2023).

2. In Figure 3, the authors suggest that the degree of prelithiation results in differences in conductivity, overpotential, and resistance. However, these differences based on Li-Si composition are already well-documented several times [Journal of Power Sources 580 (2023): 233326. / Nature Energy (2023): 1-14. / The Journal of Physical Chemistry C 115.5 (2011): 2514-2521.]. This experiment does not contribute new insights to the existing literature. Additionally, the overpotential of symmetric cell (Figure 3b) and half-cell (Figure 3e) is also influenced by the voltage of the Si active material depending on the state of charge (SOC), which is not addressed in the study. Further investigation into this phenomenon is needed.

We appreciate the reviewer's insightful comment on this matter. As the reviewer pointed out, there have been studies that demonstrated how the degree of prelithiation influence conductivity, overpotential, and resistance.

In this work, what we tried to demonstrate was to separate and quantitatively analyze electronic and ionic conductivity for the prelithiation method we chose.

Because the SOC of Si would change depending on the degree of the prelithiation, the overpotential could change as well. However, because the lithiation range of $\text{Li}_{0.25}\text{Si}$ to Li_2Si would stay as amorphous Li_xSi , we expected to see no significant difference in phase depending on prelithiation. Therefore, further studies on performance were done on only one composition of Li_1Si . As the reviewer pointed out, the overpotential can be influenced by the SOC of Si. However, Figure 3e is not full lithiation to de-lithiation but 1-hour lithiation and subsequent 1-hour delithiation at 0.2 mA cm^{-2} current density. Because all the Si used in this experiment was 5 mg ($\sim 15 \text{ mAh}$), 0.2 mAh cm^{-2} of capacity would have minimal effect on the overall SOC of Si.

We changed the main text to refer to the overpotential of the half-cell to clarify the point below.

“Figure 3e shows the half-cell configuration of Li_xSi with Li metal. All Li_xSi was first lithiated for 1-hour and then delithiated for 1-hour at the same current density. Interestingly, all Li_xSi exhibited similar overpotential during the lithiation, indicating that Li reacting with Si into Li_xSi requires a similar amount of overpotential. The non-prelithiated Si clearly showed a higher overpotential when it was first lithiated due to its poorer electronic conductivity. However, the overpotential of the delithiated process is prominently different depending on the degree of prelithiation. This indicates that the amount of prelithiation eventually affects the electronic/ionic conductivity of silicon during charge and discharge.”

3. In Figure 5, it is claimed that the ICE enhancement of full-cell by prelithiation occurs when anode-limiting. This is a reasonably known fact, but it is quite contradictory that the study uses a very high n/p ratio of 4.4, and yet the efficiency of the NCM full-cell is dominated by the cathode. Since Figure 5 only considers the case of np ratio of 1, further explanation is needed to consider the np ratio used in the actual cell.

As the reviewer pointed out, if the N/P is high, the ICE of the full cell will be dominated by anode. This was one of our main concerns, so we conducted further experiments to clarify this point. In our Si and prelithiated Si cells, the calculated N/P ratio and the practical N/P ratio differ because of the partial utilization of Si. This was proven by two experiments below in **Figure S7 and S8**. If all anode participates in the lithiation reaction, the ICE of full-cell should decrease as the N/P ratio increases. Because generally irreversible reactions happen at the relatively high reaction voltage range, most of the lithium/electron from the cathode has to be consumed by the irreversible reaction of each anode particle which results in the proportionally decrease of reversible lithiation capacity. However, our supporting experimental results show that ICEs are maintained when the N/P ratio increases from around 1 to 3.3 (**Figure S7**). So, we believe that the high N/P ratio of our full-cell doesn't affect the relationship between theoretical consideration under a realistic N/P ratio (~ 1) and our experimental result in the high N/P condition. The only reason why we use high N/P condition is the practical difficulties of fabricating the full-cell having an N/P ratio of around 1 at relatively low areal capacity (4 mAh cm^{-2}) due to the extremely high specific capacity of pure Si.

Figure S8 supports our interpretation regarding the effective N/P ratio of our full-cell. The cross-sectional FIB of N/P 1.2 and N/P 3.3 shows the stark difference, where low N/P Si has lithiated fully and the high N/P Si was partially utilized. Therefore, the N/P ratio consideration on our cell with calculated high N/P will be the same as N/P ~ 1 since only N/P ~ 1 of Si could be utilized.

The following explanations are added to the manuscript.

*“There is one more important point regarding the N/P ratio. Although the illustration in **Figure 5a** explained the ICE of full-cell depending on the cathode-/anode-limiting system based on the N/P ratio of around 1, we obtained the experimental results (**Figure 5b and 5c**) at the relatively high N/P ratio of 4.4. A high N/P ratio generally decreases ICE since irreversible lithium/electron consumption happens at a relatively high voltage (the initial stage of the lithiation process). However, our results show that the full-cell which has a wide range of N/P ratio (1~3.3) exhibits similar ICE values (**Figure S7**), because some anode parts practically don't participate in the lithiation process (**Figure S8**). It makes the effective N/P ratio of our solid-state cell around 1, consistent with the illustration. Therefore, the discussion at the beginning of this section of case 1 to case 4 is valid even with higher loading of Si.”*

Figure S7. Theoretical and experimental Coulombic efficiency of NCM-Si and LCO Si of N/P 1 to 3.3.

Figure S8. Cross-sectional FIB/SEM image of charged Si full cell of (a) N/P 1.2 and (b) 3.3. (c) EDS mapping of the charged N/P 3.3 Si cell. (d) Line scan of the charged N/P 3.3 Si cell. The line scan points are denoted in (c) with white dots.

4. In figure 6, long cycle life of full cell and efficiency data fluctuates a lot. This can also lead to reliability issues in cell evaluation.

The reviewer raised an insightful concern. However, the cell efficiency fluctuation mainly comes from the temperature fluctuation during the day. The use of a constant-temperature set-up can improve the cell performance, but we believe that the real conditions such as the temperature change during cycling also can be meaningful test conditions and give insightful results.

Reviewer #2 (Remarks to the Author):

The authors reported using Li powder to prelithiate micron size Si, which was subsequently used in a solid-state battery. Although using powder Li to prelithiate cathode e.g. hard carbon or even si is nothing new, using powder Li to lithiate Si for a solid-state battery is new. The selling point of the manuscript is using pressure to gain better prelithiation, which is no brainer. Having those said, I still think it can be published if the author can conduct more investigation to reveal detail mechanisms.

We are grateful for the reviewer's feedback and the critical points raised. We included more details on the questions raised here.

1. The kinetics of Li diffusion under different pressure. GITT of Li^+ diffusion at different pressure. Experiment and literature.

We thank the reviewer for allowing us to clarify this point. To address the kinetics of Li diffusion under different pressures, we conducted GITT of Si half-cell at a pressure range of 5 MPa to 25 MPa. The current pulse of C/20 (30 min) to relaxation (1 hour) was conducted to make sure the cell reached the equilibrium. The diffusion coefficient of Li in Si decreases at the beginning of lithiation and increases as Si gets more lithiated. Overall trend diffusion coefficients at all three pressures showed similar trends and values. From **Figure R1b and R1c**, the overpotential and internal resistance were extracted from GITT results. Initial overpotential was high due to the high energy to break crystalline Si-Si bond to form amorphous Li-Si. Therefore, we could conclude that Li-ion diffusivities in Si at the pressure range of 5 to 25 MPa are of no difference.

For much higher pressure, we could not perform GITT using Li metal counter electrode at a higher pressure range from 75 MPa, since Li metal creep during the first few biasing and the cell shorted. However, there are several molecular dynamics (MD) studies that showed the pressure effect on the diffusivity of Li in Si. When compressive stress was applied, Pan et al. concluded that diffusivity increased¹ whereas Ding et al. showed the contradicting result of the decrease in diffusivity with a higher compressive load of a few GPa level.² This discrepancy could be attributed to the complexity arising from different amorphous structures and duration times for the simulations, which could impact diffusivity calculations. Therefore, it is still premature to conclude the diffusion coefficient dependency with the presence of external high pressure of GPa level.

Figure R1. (a) Diffusion coefficients (b) Overpotential (c) Internal resistance calculated from galvanostatic intermittent titration technique (GITT) conducted at different pressure of Si half-cells

1. Pan, J. et al. Effects of stress on lithium transport in amorphous silicon electrodes for lithium-ion batteries. *Nano Energy* 13, 192–199 (2015).
2. Ding, N. et al. Determination of the diffusion coefficient of lithium ions in nano-Si. *Solid State Ionics* 180, 222–225 (2009).

2. The change of Li concentration gradient with a Si particle with time (Fick's second law may be used).

We appreciate the reviewer bringing up this matter. In the case of pure pressure-induced lithiation, the contact point between Li and Si would start to get lithiated by diffusion. When Li and Si are pressed at lower pressure, the contact area between two elements is limited due to a low degree of Li deformation. In **Figure S2a**, only the contacted part would start diffusion following Fick's second law (blue curve), whereas the non-contacted area stays with a Li concentration of 0. On the other hand, the high pressure would result in a larger contact area of Li and Si particles by more Li deformation where most of Li and Si will be in contact, therefore more Si will be lithiated. The direct evidence could be found in **Figure S2b and S2c** where cross-sectional FIB/SEM showed non-contacted voids from 200 MPa whereas more lithiation occurred at 400 MPa.

The following explanation was added in the main text with **Figure S2**.

“During the pressure-induced lithiation, Si is lithiated by diffusion of Li which would follow Fick's second law of diffusion at the contact point of Li and Si. However, the contact area of Li and Si particles is limited when the Li and Si mixture is pressed at low pressure. (Figure S2a) In Figure S2b-c, the 400 MPa pressed Li₁Si showed less remaining Li metal compared to the 200 MPa pressed pellet.”

Figure S2. (a) Schematic of pressure-induced lithiation at low pressure (top) and high pressure (bottom). The hypothetical concentration of Li with respect to distance from the Li and Si contact point is shown on the right side. (b) Cross-sectional FIB/SEM image of Li₁Si at 200 MPa (c) Cross-sectional FIB/SEM image of Li₁Si at 400 MPa.

3. Since the stabilized Li powder was used, how the residue from the Li particle impacted the Si conductivity. 45 ohm impedance in a symmetric cell seemed high (figure 3 (c)). The charge and discharge curves (Figure 3 b) also seemed strange.

We thank the reviewer for pointing this out. The electronic conductivity of lithium and silicon is around 10^7 S cm^{-1} and 10^{-6} S cm^{-1} , respectively. Therefore, the remaining Li will increase the electronic conductivity of our lithiated electrodes, which corresponds well with the DC polarization result in **Figure 3a**. The resistance of around 35 Ohm in the impedance of the symmetric cell (**Figure 3c**) mostly comes from the ionic conductivity/resistivity of the LPSCI electrolyte pellet between two electrodes ($2.2 \text{ mS cm}^{-1} = 32 \text{ Ohm}$ in our pellet cell configuration). In **Figure 3c**, the reason why the $\text{Li}_{0.25}\text{Si}$ symmetric cell shows a relatively higher resistance value than the other two cells is the poor contact between the LPSCI electrolyte pellet and the electrode due to the stiffness of low lithiated silicon. In addition, we believe that the slope shape of the lithiation/delithiation profile in **Figure 3b** is due to the high overpotential of delithiation reaction from the lithiated silicon. Since all reactions on this symmetric cell must undergo the delithiation reaction of lithiated silicon (lithiation process ($< 0 \text{ V}$ (vs. Li/Li^+)) on working electrode = delithiation process on counter electrode, delithiation process ($> 0 \text{ V}$ (vs. Li/Li^+)) on working electrode = lithiation process on counter electrode), high overpotential of delithiation reaction affect the large polarization and thus more sloppy profile shape on the cell reaction.

The following are added to the manuscript.

“Before plating and stripping, the resistance of $\text{Li}_{0.25}\text{Si}$ is higher than that of Li_1Si or Li_2Si due to the poor contact between two electrodes and LPSCI electrolyte pellet because of stiffness of low lithiated silicon (please note that the resistance of around 35 Ohm mainly comes from the ionic conductivity of LPSCI pellet between two electrode ($\sim 2.2 \text{ mS cm}^{-1}$)).”

4. It is strange that the authors use either LCO and NMC811. Please provide the rationale regarding how the full cell was balanced.

The rationale behind our choice of LCO or NMC is because of the initial Coulombic efficiency difference, thus it would be a suitable cathode material to explain cathode and anode limiting in full-cells. As can be seen from Figure S3, the ICE of NCM is significantly lower than that of LCO, so we thought that these two cathodes could be a good comparison to demonstrate the effect of the ICE difference. The first cycle irreversible capacity of layer oxide cathode has been studied in previous research as follows. Professor Stanley Whittingham’s group explained very clearly that the first cycle irreversible capacity of NCM811 mainly comes from the slow lithium kinetics and partly from surface change and cathode electrolyte interface formation.¹ For LCO, as long as the potential does not exceed 4.35 V (vs. Li/Li^+), almost all capacity could be back during the first cycle discharge, free from unstable structural change at high potential.^{2,3} In the full cell, since NCM and LCO have different specific capacities (mAh g^{-1}), we balanced the amount of cathode in terms of areal capacity (mAh cm^{-2}). In addition, we also choose the charge capacity as a base on the calculation of the amount of cathode, so that both two full-cells show the same charge capacity in terms of areal capacity which is around 4 mAh cm^{-2} .

1. Zhou, H., Xin, F., Pei, B. & Whittingham, M. S. What Limits the Capacity of Layered Oxide Cathodes in Lithium Batteries? *ACS Energy Lett.* 4, 1902–1906 (2019).
2. Li, J. et al. Structural origin of the high-voltage instability of lithium cobalt oxide. *Nat. Nanotechnol.* 16, 599–605 (2021).
3. Amatucci, G. G., Tarascon, J. M. & Klein, L. C. CoO_2 , The End Member of the Li_xCoO_2 Solid Solution. *J. Electrochem. Soc.* 143, 1114–1123 (1996).

5. Please provide more detail analysis on impedance data, e.g. whether the data is in compliance with linearity? Separation of different electrochemical processes.

We thank the reviewer’s comment on the impedance data. The ionic conductivity of LPSCI is calculated from the Nyquist plot having an x-intercept of 32 Ohm yielding the conductivity of 2.2 mS cm^{-1} , which corresponds with the ionic conductivity of LPSCI we are used in this work. In the low-frequency range, the Nyquist plot showed around 45 degrees for all Li_xSi impedance measurements indicating diffusion of charge resulting in Warburg impedance. This could be attributed to the solid-state diffusion of Li in Si in the Li_xSi non-blocking electrode. Because of the lack of a semi-circle in the measured data, further fitting was not conducted. For the

resistance value of each species, the trend of higher resistance for $\text{Li}_{0.25}\text{Si}$ compared to more lithiated Si would likely be from having less efficient contact between LPSCI and lithiated Si.

Figure R2. Nyquist plot of LPSCI pellet and $\text{Li}_1\text{Si}/\text{LPSCI}/\text{Li}_1\text{Si}$

Reviewer #3 (Remarks to the Author):

In this paper, an electrochemical prelithiation strategy for Si anode was employed to improve the low initial Coulombic efficiency (ICE) and conductivity of all-solid-state batteries. Using a Li₁Si anode and a LCO cathode, the full cell achieved over 95% of ICE. Furthermore, the capacity retention with Li₁Si anode maintained 73.8% after 1000 cycles, which was 15% higher than that of the pure Si anode. A high capacity of up to 10 mAh cm⁻² was obtained using Li₁Si and dry-processed LCO cathode film, indicating that the prelithiation method may be suitable for high loading next-generation all-solid-state batteries. The pressure-induced prelithiation strategy can effectively improve the capacity retention and energy density. However, a few concerns need to be addressed before published. I recommend a major revision for the manuscript.

We are grateful for the reviewer's feedback and the critical points which were raised. We included more details on the questions raised here. Detailed responses will be discussed below.

1. Due to the large volume expansion and the "solid-solid" contact at the interface, the interfacial structure would be greatly influenced. Prof. Meng reported in an earlier work published in Science that: "the interfacial contact area between the SSE and the mSi electrode is reduced to a two-dimensional (2D) plane. After lithiation of μ Si, the 2D plane is retained despite volume expansion, preventing the generation of new interfaces." In this study, the interfacial structures between Li₁Si and solid electrolytes may need to be presented as well, a few SEM images are recommended to provide for comparison.

We appreciate the reviewer to help us clarify this point. We added **Figure S9** to show a cross-sectional FIB/SEM image of pristine, charged, and discharged Li₁Si. The discussion on morphology and interface was added as follows.

*"The morphology of Li₁Si upon charging and discharging was demonstrated in **Figure S9**. The charged Li₁Si cross-sectional image shows that the partial utilization of Si is valid even in the Li₁Si case, meaning only the LPSCl facing side of Li₁Si gets lithiated whereas the opposite current collector side still has unreacted Li metal (dark) as we discussed earlier. The discharged sample exhibited surface cracks in some parts of the electrode (**Figure S9f**) where the cross-sectional image of the non-cracked part was shown in **Figure S9d** and the cracked part shown in **Figure S9e**, which indicated the existence of a huge volume change of the silicon electrode."*

Figure S9. Cross-sectional FIB/SEM image of (a) pristine, (b) charged (d) discharged non-cracked spot (e) discharged cracked spot. Surface SEM image of (c) charged and (f) discharged. All images were obtained from Li₁Si samples. The charged and discharged samples were all first cycle results of Li₁Si cells.

2. The prelithiation process makes the Li_1Si phase as the starting composition for the subsequent electrochemical cycles (charging \leftrightarrow discharging ...). However, it is well known that the lithiated $\text{c-Li}_{3.75}\text{Si}$ phase (the charged state) would transform back to the a-Si phase during a discharging process, which may result in a fact that the prelithiated Li_1Si phase cannot be recovered in the subsequent cycles. The authors may need to address this issue, which relates to the mechanistic origin of the battery performance improvement.

We totally agree with the reviewer's comment. If the operating condition of the effective N/P ratio of our prelithiated silicon full-cell is around 1 based on the charge capacity (please see the comments from other reviewers or our modified manuscript), we believe that the anode part that participates in the lithiation reaction can uptake the lithium/electron up to a $\text{c-Li}_{3.75}\text{Si}$ phase, and release around 95% of lithium/electron during delithiation in our LCO/ Li_1Si cell (since the ICE of this full-cell is around 95%). It means that the anode should be discharged up to $\text{a-Li}_x\text{Si}$ whose x value should be very small compared to 1. Thus, the charged/discharged state of the silicon anode on the subsequent cycle should be $\text{c-Li}_{3.75}\text{Si}$ and $\text{a-Li}_x\text{Si}$ ($x \ll 1$), which is consistent with the reviewer's point of view. The main reason for prelithiation is the improvement of the ICE of silicon full-cell. In the case of ordinary silicon which doesn't have an additional lithium source in the pristine state, it cannot be delithiated (discharged) up to the $\text{a-Li}_x\text{Si}$ ($x \ll 1$, the discharged state of LCO/ Li_1Si cell) because some of the lithium is consumed or trapped irreversibly, which results in the low ICE of the full-cell. Therefore, the main effect of prelithiation is the increase of the amount of lithium (and thus electron) which can be transferred between cathode and anode (and thus cell capacity) by introducing the additional lithium/electron source which can compensate some amount of lithium/electron loss from irreversible reaction or trapping. In addition, this prelithiation enlarges the range of SOC of silicon (higher SOC of lithiation state and lower SOC of delithiation state) which results in improved capacity and cycle performance.

3. In Figure 2b, the coexistence of Li-Si , Si , and Li phases is observed after the prelithiation process. However, it remains unclear whether the residual Si atoms in Figure 2b will engage in the electrochemical cycling process, potentially resulting in increased volume expansion. We suggest the authors explain this question in revision.

We appreciate the reviewer's comment to clarify this point. In terms of thermodynamics aspect, lithium metal can react with silicon (lithium metal can give lithium-ions and electrons to the silicon by reducing silicon), which means the reaction between lithium metal and silicon should be spontaneous. However, as all electrochemical reaction does, it needs overpotential (more energy kinetically) to make the reaction happen. In our prelithiated electrode, the additional energy was the lithiation (charging process) of the Li_xSi electrode. Additional energy provided during the charging process makes the overcome of activation kinetic energy of the lithium-silicon alloying reaction, so it initiates this alloying reaction. After that, now this prelithiated lithium/electron can participate in subsequent electrochemical reactions. In addition, volume expansion is indeed crucial to the part when discussing alloy anodes. We marked the Li_1Si layer thickness in **Figure S9** and the following discussion has been added to the manuscript.

"The volume expansion from pristine to charged state was ~200% (Figure S9), and the discharged state showed minimal difference in thickness compared to the pristine state. The volume expansion rate seems to be below the reported lithiated Si , but this is mostly because Li_1Si was partially lithiated where part of the anode was not utilized. Still, the relatively lower volume expansion rate could benefit the long-term cycling of Li_1Si ."

4. Figure 5a appears to be quite small, making it difficult to read. We suggest authors to consider rearranging the layout.

We thank the reviewer's suggestion. The following change has been made to make the text clearer.

Figure 5. a) Schematic illustrating ICE estimates of the Si and Li₁Si paired with NCM and LCO cathodes. First-cycle voltage profiles of b) NCM811 and c) LCO paired with Si and Li₁Si at C/20.

5. A minor typo: in line 330, there should be a space between “10” and the unit “mAh”

We thank the reviewer for pointing out the error. The following change has been made.

“Furthermore, using Li₁Si, a high areal capacity of 10 mAh cm⁻² was achieved and demonstrated using a dry-process LCO film, demonstrating that the lithiated Si could be a suitable candidate to be used in high-energy-density next-generation batteries.”

Reviewer #4 (Remarks to the Author):

The manuscript reports the preparation of a lithiated Si electrode and its investigation in various cell configurations using Li-argyrodite as electrolyte material. The authors demonstrate that in ASSB the use of prelithiated Si electrodes improves the cycling performance, especially if the capacity of the anode was limiting. Impressive cycling performances as well as higher CAM loading through dry processing techniques are presented by the authors. This manuscript can be accepted after addressing the following issues.

We appreciate the reviewer's positive comments on our work. The points raised by reviewers were discussed in the following.

1. In Figure 1 presents the spider diagram of the characteristic properties of Si over prelithiated Si for ASSBs. The color choice makes it difficult to differentiate between the Si and prelithiated Si electrodes. The Initial CE is linked with the formation of the passivation layer on Si particles. Why in the case of Si and prelithiated Si are no differences observed? Furthermore, using Si and prelithiated Si just a small increase is observed for the energy density and in the manuscript text only ICE, electronic conductivity and Li diffusion are mentioned. Can you please comment why energy density is not mentioned or discussed, as based on the material characteristic properties some hypothetical calculations can be made?

We appreciate the reviewer's concern about **Figure 1** and energy density. The initial Coulombic efficiency has been reported to be mainly governed by SEI formation on the surface of Si particles in the previous research. However, as reported in our previous report, the SEI formation only occurs at the apparent contact area between the LPSCI electrolyte pellet and Si anode since our pure Si anode doesn't contain the electrolyte inside the anode, irreversible capacity loss from SEI formation can be minimized. Because this apparent contact area should be the same in the case of Si and prelithiated Si, the amount of irreversible capacity loss from the SEI formation should be the same. The reason why we can obtain high ICE on prelithiated Si is due to the existence of an additional lithium/electron source in the anode which can be released during discharge (delithiation).

The color scheme of **Figure 1** was changed to better distinguish the difference between the two cases as below.

The energy density based on the assumption that we have a thin solid electrolyte has been calculated and added in the main text as follows.

"The energy density has been calculated to be 236 Wh kg⁻¹ and 947 Wh L⁻¹ for the high loading 10 mAh cm⁻² cell shown in Figure S11, which is based on the assumption that the solid electrolyte layer of 30 μm."

Figure 1. A radar comparison chart of Si (light blue shade) and prelithiated Si (green shade) anodes for various electrochemical properties and battery performance metrics.

2. In the introduction part, the focus is made on the pre-lithiation method. However, in the case of Si and solid-state electrolytes the discussion is not mentioning with too many details the effect of pre-lithiation on the material polarizability. Can you please comment if the prelithiation, in this case, is made exclusively in order to improve the Li conductivity in the high Si-loading anode material?

We believe that one of the main advantages of prelithiation is that we can engineer the range of silicon SOC during charge and discharge. Since we intentionally add additional lithium/electron on the silicon at the pristine state, the initial SOC before charging is already higher than 0 (slightly lithiated state). Compared with pure Si which has low electronic conductivity (it is close to an insulator without any doping), and no ability to conduct lithium-ion (because of no lithium source), lithiated silicon has quite high electronic conductivity (doping effect) and lithium-ion conductivity which can improve the kinetics and reduce polarization, which can be observed in **Figure 2** results. As reported in our previous report about pure Si ASSB on Science, after lithiating the silicon, it can deliver the lithium ion to the bulk part of the silicon anode (that was the main mechanism of how pure silicon without solid-state electrolyte can react as the anode material). In addition, since it is known that silicon anode with a high SOC state shows higher electronic/ionic conductivity due to the reason mentioned above, it is very beneficial to make the range of silicon SOC range deeper by prelithiation. In summary, we'd like to suggest the rationale that the prelithiation can improve ICE due to the additional lithium/electron source as well as improved average ionic/electronic conductivity during charge and discharge.

3. Figure 2b and Figure 2c indicate the evolution of the electrode morphology and the change of Li metal environment from SLMP to prelithiated Li-Si phase. Can you comment on the homogeneity of the assigned-Li₁₃Si₄ phase throughout the electrode bulk after applying pressure from the ss-NMR perspective? Are there any resonance shifts of the Li-Si regions that can be assigned to other Li-rich-Si phases that are formed during the 30s or 3 min applied pressure, and later evolving into Li₁₃Si₄ phase?

We thank the reviewer for this question. When the Li₁₃Si₄ mixtures are pressed, the formed Li-Si alloy is likely composed of more than one Li_xSi_y phase but the broad resonances observed in the NMR spectra obtained on the corresponding samples (shown in Figure 2 of the manuscript) are difficult to resolve into individual constituent resonances as the measurements were run in static mode to avoid the evolution of the sample induced by frictional heating from sample spinning. As a result, precise identification of the phases present in each sample is difficult and subject to error.

Nevertheless, the following explanation and the figure were added to the Supporting Information.

*“The homogeneity of NMR spectra with the fitting is provided in **Figure S4**”*

“We conducted additional fits on each of the spectra presented in Figure 2c of the manuscript. Fits are presented in Figure 1 below. The Li-Si resonance for the unpressed sample is centered at 16.5 ppm, which corresponds well to the reported resonance for Li₇Si₃.¹ The spectrum obtained on the sample pressed at 100 MPa for 30s exhibits resonances at 69.4 ppm and 3 ppm, which are tentatively assigned to Li₂₁Si₅ and Li₁₅Si₄, respectively.¹ Fits for the spectra obtained on the samples pressed at 200 MPa and 400 MPa for 30s as well as 200 MPa for 3 min are nearly identical and are fit with a single sharp resonance centered between 10.2-11.5 ppm, and a much broader resonance centered between 15-30 ppm. The sharp resonance in these samples is tentatively assigned to Li₁₃Si₄, while the broad component is due to a combination of other Li_xSi_y phases present.¹ An additional, weak signal at -67 ppm is observed in each spectrum and could not be assigned to a known Li-Si phase.

Based on the above results, it appears that the least homogenous sample is the sample pressed at 100 MPa for 30s as it likely contains both Li₂₁Si₅ and Li₁₅Si₄. While the unpressed sample spontaneously formed some Li₇Si₃, it is still mostly composed of unincorporated Li metal. Pressing the Li-Si mixtures at higher pressures and for longer durations forms a larger phase fraction of the Li-Si alloy and appears to yield a more consistent mixture of Li_xSi_y phases, mostly consisting of Li₁₃Si₄. While consistency across samples should not be conflated with a homogenous distribution of phases throughout individual samples (NMR does not provide any information on the spatial distribution of the phases), the larger pressure applied on these samples likely forces more intimate contact and reactions between the SLMP and Si alloy that drives the formation of mostly Li₁₃Si₄.”

1. Key, B. et al. Real-Time NMR Investigations of Structural Changes in Silicon Electrodes for Lithium-Ion Batteries. *J. Am. Chem. Soc.* 131, 9239–9249 (2009).

Figure S4: Fits conducted on ^7Li ssNMR spectra obtained on Li_xSi mixtures pressed under various conditions. All spectra were obtained at 18.8 T with a spin-echo pulse sequence using 30° and 60° flip angles under static conditions.

- At page 5, the authors mention that the Li-Si alloy phases can be quantified from the Knight shift around 0ppm. Is the 0ppm shift not characteristic to Li in the SEI while Li-Si phases, according to Key et al., *J. Am. Chem. Soc.* 2009, 131, 26, 9239–9249, should be around 18- to 6ppm? Can you comment?

In the experiments described on Page 5 and **Figure 2** of the manuscript, mixtures of Li and Si are pressed at various times and pressures and later characterized with SEM and NMR. None of the corresponding samples are incorporated into an electrochemical cell before the characterization and therefore, do not have an SEI component. In **Figure 2** of the manuscript, the broad resonances obtained on the pressed samples assigned to the Li-Si phases are centered between 8-15 ppm and fall well within the range of ppm values described for Li-Si phases by Key et al.¹ The Li-Si resonances likely consist of a combination of resonances corresponding to multiple Li-Si phases within the sample. Potential constituent phases of the Li-Si alloys formed from pressing include $\text{Li}_{15}\text{Si}_4$ (6.0 ppm), $\text{Li}_{13}\text{Si}_4$ (11.5 ppm), Li_7Si_3 (16.5 ppm), and $\text{Li}_{12}\text{Si}_7$ (18.5 ppm).¹ While the identification of individual resonances corresponding to distinct Li-Si phases is made difficult by the lack of resolution from conducting the NMR measurements statically to avoid heating effects from spinning, further attempts at phase identification are provided in the response to Question 3.

1. Key, B. *et al.* Real-Time NMR Investigations of Structural Changes in Silicon Electrodes for Lithium-Ion Batteries. *J. Am. Chem. Soc.* **131**, 9239–9249 (2009).

- In Figure 3a, the addition of 0.25 equivalents of Li already increases the electric conductivity to almost two orders of magnitude compared to pure Si electrode. The effect is also observable in Figure 3e, where it seems that the overpotential decreases to negative values (Li plating?) before some uptake of Li is occurring. Can you please comment?

We thank the reviewer for allowing us to clarify this point. In Figure 3e, only Si decreased to negative values initially during the lithiation of Li_xSi . This is attributed to the high resistance of the Si electrode which leads to initial Li plating on Si whereas lithiated Si did not have this high overpotential leading to Li plating behavior.

6. At page 7, the following statement is made: “This indicates that it is harder to delithiate Li_xSi depending on how much Li is in Si.” How this statement aligns with the behavior of $\text{Si}|\text{LPSCl}|\text{Li}$ cells?

We thank the reviewer’s comment on this point. In Figure S5c, $\text{Si}|\text{LPSCl}|\text{Li}$ half-cell data was demonstrated, where Si was first lithiated and then delithiated. During the delithiation, the potential drastically increased indicating more energy was needed to delithiated Si at the later stage of delithiation.

7. In the next sentence : “In addition, the irreversible capacity from the initial lithiation and delithiation was observed in Li_xSi paired with Li counter electrode.” the discussion is made upon the ICE or CE of the Li_xSi paired with Li. The value of the ICE or CE of the $\text{Li}_1\text{Si}|\text{LPSCl}|\text{Li}$ should be provided in the text (also for the sake of the discussion in subsection 2.3).

We appreciate the reviewer’s comment on this matter. We agree that to complete the story, Li_1Si half-cell with Li metal counter electrode data would be needed. However, the minimum loading we could make for Li_1Si was $\sim 10 \text{ mAh cm}^{-2}$, which ended up in the cell shorting during the delithiation stage. Therefore, in the main text, instead of full lithiation – delithiation, we conducted 1-hr lithiation and then 1-hr delithiation to demonstrate the electronic and ionic conductivity of Li_xSi .

8. In Figure 4 only the initial cycle is presented. This should be clearly indicated also in the manuscript text. The Axis label ICE (Figure 4b) should be corrected to “Initial”.

We appreciate the reviewer’s comment to make this point clear. The following changes have been made to the main text.

“In Figure 4a, full cells with the following configuration, $\text{Li}_x\text{Si} | \text{LPSCl} | \text{LCO}$, were fabricated and cycled at C/20 to study the effect of various prelithiation amounts in Si, which was to evaluate the first cycle performance with limited lithium inventory.”

“Figure 4. b) Initial Coulombic efficiency trend of Li_xSi ($x = 0, 0.25, 1, \text{ and } 2$)”

9. The title of the subsection 2.3 should report the N/P ratio. The authors use NCM811/LCO or $\text{Si}/\text{Li}_1\text{Si}$ that is confusing. What is the areal capacity of the Si or Li_1Si electrodes?

We thank the reviewer’s comment to clarify this point. The N/P ratio of each cell has been calculated based on the actual capacity we achieved from the half cells of NCM, LCO, and Si with the Li metal counter. Please note that all electrode areas are the same with a value of 0.785 cm^2 . We calculated the charge capacity of NCM (200 mAh g^{-1}) and LCO (150 mAh g^{-1}). The cathode areal loading was controlled to be 4 mAh cm^{-2} for both NCM and LCO. For the lithiation capacity of Si, we used the value of 2800 mAh g^{-1} because, in the practical condition of C/20, we could not get to the capacity of 3580 mAh g^{-1} ($\text{Li}_{15}\text{Si}_4$), instead, we could achieve 2800 mAh g^{-1} . The loading of the anode side was controlled to be 5 mg of Si, which yielded the areal capacity of 17.8 mAh cm^{-2} for Si and 14.6 mAh cm^{-2} for Li_1Si . Therefore, the N/P ratios of NCM811/Si and LCO/Si cell are both 4.4. The N/P ratios of NCM811/ Li_1Si and LCO/ Li_1Si are both 3.7.

We also added the N/P ratio consideration in subsection 2.3 and added the practical N/P ratio in the solid-state cell setup as follows.

*“There is one more important point regarding the N/P ratio. Although the illustration in **Figure 5a** explained the ICE of full-cell depending on the cathode-/anode-limiting system based on the N/P ratio of around 1, we obtained the experimental results (Figure 5b and 5c) at the relatively high N/P ratio of 4.4. A high N/P ratio generally decreases ICE since irreversible lithium/electron consumption happens at a relatively high voltage (the initial stage of the lithiation process). However, our results show that the full-cell which has a wide range of N/P ratio (1~3.3) exhibits similar ICE values (**Figure S7**), because some anode parts practically don’t participate in the lithiation process (**Figure S8**). It makes the effective N/P ratio of our solid-state cell around 1, consistent with the illustration. Therefore, the discussion at the beginning of this section of case 1 to case 4 is valid even with higher loading of Si.”*

Figure S7. Theoretical and experimental Coulombic efficiency of NCM-Si and LCO Si of N/P 1 to 3.3.

Figure S8. Cross-sectional FIB/SEM image of charged Si full cell of (a) N/P 1.2 and (b) 3.3. (c) EDS mapping of the charged N/P 3.3 Si cell. (d) Line scan of the charged N/P 3.3 Si cell. The line scan points are denoted in (c) with white dots.

- In subsection 2.3, the notation Li_xSi should be referred to the composition of the negative electrode, i.e. Li_1Si (according to the results, only the phase Li_1Si and Si electrodes were used). In subsection 2.2 it is appropriate to use Li_xSi .

We thank the reviewer for commenting on the correction of the appropriate notation. We changed the notation from Li_xSi to Li_1Si in subsection 2.3 as we discussed only Li_1Si for the lithiated Si, and we kept the Li_xSi in subsection 2.2 part since we covered the wide range of lithiation of Si in this section.

“Two different cathodes were paired with Si and Li₁Si to elucidate the limiting component of the system (Figure 5).”

- Figure 3e and Figure S3 show the half-cell data of Si|LPSCI|Li cells. Why the evolution of the voltage profile is different? Also, in the caption 10MPa are mentioned to be used for the applied pressure while 15MPa are indicated in the Figure S3. Please correct.

We appreciate the reviewer pointing out this mistake. The following correction has been made to reflect the comment.

Figure S5. Half-cell data of (a) NCM, (b) LCO, and (c) Si with Li metal counter electrode. All cells were cycled at C/10, room temperature, and 10 MPa.

- The case 4 in Figure 5a is not discussed in the text. Since this is the outcome of the manuscript is based on this effect, the authors should further support the schematic with significant ICE values and discussion in the subsection 2.3.

We appreciate the reviewer’s comment on this. The following statements have been added to the main text to explain case 4 better in subsection 2.3.

“However, the ICE of LCO cells increased significantly from 78.3% to 95.7% (Figure 5c). The first-cycle voltage profiles from these cells were consistent with the hypothesis illustrated in Figure 5a Case 2 and Case 4. As Figure 5a Case 4 achieved the highest ICE of 95.7%, further long cycling and higher loading efforts are all made in this configuration.”

- In Figure 6c, the discharge capacity decay seems to be steeper for the Li₁Si cell, rather than for Si. The authors explain the increase during the initial <20 cycles for the Li₁Si cell as a result of the residual Li present in these electrodes. Why in the cycling stability plots of both materials, such increase and decrease occur for both samples?

We appreciate the reviewer’s concern about the long cycling data. The fluctuation in the long cycling data for both cells came from the temperature fluctuation during the day. The difference between the two cells is that the discharge capacity of Li₁Si increased slightly because of the residual Li reaction, whereas Si does not have a noticeable initial increase, rather constant decay in discharge capacity.

- Are the values for the current density used in Figure 3e and Figure 6c similar?

We thank the reviewer for bringing this matter up. The current densities in Figure 3e and Figure 6c are 0.2 mA cm⁻² and 5 mA cm⁻², respectively. With that said, the current densities in the two figures are significantly different. The reasons why we chose these current densities are as follows. For Figure 3e, we wanted to examine how reversible lithiated Li_xSi with minimized kinetic effect. Therefore, the very low current density was chosen to minimize the potential kinetic limitation. For Figure 6c, we wanted to examine at a practically high rate to test the practicality of the cell.

15. In subsection 2.3 the authors concluded that the prelithiation is only effective if a limiting anode is used. How is this statement supportive to the last paragraph of the manuscript?

We appreciate the reviewer's comment. The last paragraph of the manuscript is the part regarding the higher loading of LCO with the fixed amount of Li_1Si . To make it clearer, we conducted more experiments on N/P ratio consideration and added the content in the last part of section 2.3. From these results, we could conclude that a high N/P ratio would end up with the same initial Coulombic efficiency because of the partial utilization of Si. Since even the highest loading of LCO cell still exhibits an N/P ratio slightly higher than 1, Li_1Si would be partially utilized, which gives similar practical N/P ratios for all cells. With that being said, the last paragraph's result supports subsection 2.3.

16. In the last part of the manuscript, the authors present the effect of higher active material loading for the cathode material. It is remarkable that the authors achieve such high loading. The authors should present and briefly discuss the effect of such high mass loadings on the kinetics of the cathode material as well as on the long cycling performance.

We appreciate that the reviewer raised an important point to discuss. We were also surprised that this high-loading thick cathode worked in our system. We chose the dry processing of thick cathode based on the result showing the dry electrode approach facilitates a more homogeneous electrochemical pathway⁴², the film consists of cathode, catholyte, and PTFE binder. Inhomogeneous reaction within the thick electrode has been reported previously, showing lithium ion diffusion limitation⁴⁰ and the resulting state of charge variation.⁴¹ Therefore, we conducted the cycling at the low rate of C/20 to minimize the kinetic issue. A slight decrease in utilization was observed in the high-loading cells compared to the small-loading cells.

The following texts have been added to the main text.

“As such, increasing the cathode loading to match the high capacity of anode was needed. However, with regards to the high-loading thick electrode, an inhomogeneous reaction within the thick electrode has been reported previously, showing lithium-ion diffusion limitation which resulted in the state of charge variation.^{40,41} Therefore, dry processing of cathode film consisting of cathode, catholyte, and polytetrafluoroethylene (PTFE) binder was fabricated to achieve a better homogeneous electrochemical pathway within the thick electrode.⁴²”

40. Park, K.-Y. et al. Understanding capacity fading mechanism of thick electrodes for lithium-ion rechargeable batteries. *Journal of Power Sources* 468, 228369 (2020).

41. Kim, H. et al. Failure mode of thick cathodes for Li-ion batteries: Variation of state-of-charge along the electrode thickness direction. *Electrochimica Acta* 370, 137743 (2021).

42. Yao, W. et al. A 5 V-class cobalt-free battery cathode with high loading enabled by dry coating. *Energy Environ. Sci.* 16, 1620–1630 (2023).

REVIEWER COMMENTS

Reviewer #1 (Remarks to the Author):

I have carefully reviewed the authors' response. I acknowledge that the given approaches and results might have a certain impact on the ASSBs. However, I am still negative on the novelty of the current work because the main themes and ideas of the current work is a simple combination of the existing knowledge and processes. For example, prelithiation of Si in the presence of metallic Li by applying pressure is well known, and the current work simply adopted the given scheme in the pressuring step in the ASSB cell fabrication. Seriously, a similar prelithiation scheme for ASSBs with Si anodes has been reported (<https://doi.org/10.1016/j.etsr.2023.100277>). Prelithiation onto bare Si without carbon and other additives is also well known like the authors have also published (DOI: 10.1126/science.abg7217).

Reviewer #2 (Remarks to the Author):

The authors have made efforts to address the technical queries raised by the reviewers. I find most of the specific concerns were adequately addressed. However, I maintain that the concerns expressed by reviewer 1 and 2 regarding novelty still remain. The manuscript could be published if the authors can explicitly articulate and define the novel contributions, which is unclear at the current form.

Reviewer #3 (Remarks to the Author):

The authors have addressed all of my concerns. I recommend this work to be published in Nature Communications.

Reviewer #4 (Remarks to the Author):

I consider that the authors did quite a good job by considering all the concerns raised.

REVIEWER COMMENTS

Reviewer #1 (Remarks to the Author):

I have carefully reviewed the authors' response. I acknowledge that the given approaches and results might have a certain impact on the ASSBs. However, I am still negative on the novelty of the current work because the main themes and ideas of the current work is a simple combination of the existing knowledge and processes. For example, prelithiation of Si in the presence of metallic Li by applying pressure is well known, and the current work simply adopted the given scheme in the pressuring step in the ASSB cell fabrication. Seriously, a similar prelithiation scheme for ASSBs with Si anodes has been reported (<https://doi.org/10.1016/j.etsr.2023.100277>). Prelithiation onto bare Si without carbon and other additives is also well known like the authors have also published (DOI: 10.1126/science.abg7217).

We appreciate the reviewer's constructive feedback. The first paper that the reviewer mentioned was published after we submitted this work to Nature Communications (August 2023). The second paper published in Science covered Si without carbon and additives, but this work did not prelithiate the Si, yielding low ICE.

Although prelithiation using Li metal powder is a well-known approach, with the help of the fabrication process of an ASSB which is the pressurization step, we achieve the effect of prelithiation with minimum extra steps, and we believe that this approach hasn't been reported yet. In addition, this work also has value in the sense that we analyzed the effect of prelithiation in ASSB depending on the type of cathode active material and the cell design parameter including the N/P ratio. More detail on the effect of N/P ratio which has not been reported before, we showed that high N/P ratio Si cells behaved completely different from the liquid counterparts with the presence of excess Si. That is, instead of having a low state of charge within the anode, Si was partially lithiated and acted like a N/P ratio of 1 cell. This behavior significantly influenced the ICE of the full cell, which was discussed in the main text. Moreover, the improved ICE was achieved even with a high loading of 10 mAh cm⁻² from the prelithiated Si, which achievement in terms of areal capacity is the highest value for our best knowledge, and we believe that it shows the true viability of the Si anode with a high loading cathode.

We believe our work benefits understanding prelithiated Si for ASSB, and we have added the paragraph below to the Introduction to show the value and novelty of our work.

"In this work, the effectiveness of the prelithiation in ASSB was assessed depending on cathode selection and N/P ratio for the first time. Regarding long term cyclability, a cell of prelithiated Si paired with LCO showed a high ICE of over 95% with a stable cyclability for 1000 cycles at 5 mA cm⁻² current density.

Interestingly, we revealed that cathode irreversibility determined the effect of prelithiation on the full-cell and high N/P ratio Si cells behaved completely different from the liquid counterparts with the presence of excess Si. For solid-state cells, instead of having a low state of charge within the anode, Si becomes partially lithiated at its 2D interface and consistently acts like a cell with N/P ratio of 1. This behavior can be translated within a full cell, where the ICE was constant regardless of the N/P ratio. Moreover, the improved ICE was achieved even with a high loading of 10 mAh cm⁻² from the prelithiated Si, showing the true viability of the Si anode with a high-loading cathode. Based on the novel understanding, our work provides the insight to properly adopt prelithiated Si in ASSB configuration"

Reviewer #2 (Remarks to the Author):

The authors have made efforts to address the technical queries raised by the reviewers. I find most of the specific concerns were adequately addressed. However, I maintain that the concerns expressed by reviewer 1 and 2 regarding novelty still remain. The manuscript could be published if the authors can explicitly articulate and define the novel contributions, which is unclear at the current form.

We thank the reviewer's comments on our previous response. We tried to articulate the novelty of our work in the introduction part as follows.

“In this work, the effectiveness of the prelithiation in ASSB was assessed depending on cathode selection and N/P ratio for the first time. Regarding long term cyclability, a cell of prelithiated Si paired with LCO showed a high ICE of over 95% with a stable cyclability for 1000 cycles at 5 mA cm⁻² current density.

Interestingly, we revealed that cathode irreversibility determined the effect of prelithiation on the full-cell and high N/P ratio Si cells behaved completely different from the liquid counterparts with the presence of excess Si. For solid-state cells, instead of having a low state of charge within the anode, Si becomes partially lithiated at its 2D interface and consistently acts like a cell with N/P ratio of 1. This behavior can be translated within a full cell, where the ICE was constant regardless of the N/P ratio. Moreover, the improved ICE was achieved even with a high loading of 10 mAh cm⁻² from the prelithiated Si, showing the true viability of the Si anode with a high-loading cathode. Based on the novel understanding, our work provides the insight to properly adopt prelithiated Si in ASSB configuration”

Reviewer #3 (Remarks to the Author):

The authors have addressed all of my concerns. I recommend this work to be published in Nature Communications.

We are grateful for the reviewer’s feedback and approval for publishing.

Reviewer #4 (Remarks to the Author):

I consider that the authors did quite a good job by considering all the concerns raised.

We appreciate the reviewer’s positive comments on our work.